# MITIGATING SPURIOUS BIAS WITH LAST-LAYER SELECTIVE ACTIVATION RETRAINING

## ABSTRACT

Deep neural networks trained with standard empirical risk minimization (ERM) tend to exploit the spurious correlations between non-essential features and classes for predictions. For example, models might identify an object using its frequently co-occurring background, leading to poor performance on data lacking the correlation. Last-layer retraining approaches the problem of over-reliance on spurious correlations by adjusting the weights of the final classification layer. The success of this technique provides an appealing alternative to the problem by focusing on the improper weighting on neuron activations developed during training. However, annotations on spurious correlations are needed to guide the weight adjustment. In this paper, for the first time, we demonstrate theoretically that neuron activations, coupled with their final prediction outcomes, provide self-identifying information on whether the neurons are affected by spurious bias. Using this information, we propose last-layer selective activation retraining (LaSAR), which retrains the last classification layer while selectively blocking neurons that are identified as spurious. In this way, we promote the model to discover robust decision rules beyond spurious correlations. Our method works in a classic ERM training setting where no additional annotations beyond class labels are available, making it a practical and efficient post-hoc tool for improving a model's robustness to spurious correlations. We theoretically show that LaSAR brings a model closer to the unbiased one and empirically demonstrate that our method is effective with different model architectures and can effectively mitigate spurious bias on different data modalities without requiring annotations of spurious correlations in data.

## 1 INTRODUCTION

Deep neural networks trained with empirical risk minimization (ERM) tend to develop *spurious bias* — a tendency to use spurious correlations for predictions. A spurious correlation is a non-causal correlation between a class and a feature non-essential to the class, called a spurious feature. For example, waterbird and water background may form a spurious correlation (Sagawa et al., 2019) in waterbird predictions: a water background feature is non-essential to the waterbird class, even though there are 95% images of waterbird (Fig. 1) with water backgrounds. In contrast, a core feature such as bird feathers causally determines a class. A model with spurious bias may still achieve a high prediction accuracy (Beery et al., 2018; Geirhos et al., 2019; 2020; Xiao et al., 2021) even without core features, such as identifying an object only by its frequently co-occurring background (Geirhos et al., 2020). However, the model may perform poorly on the data where spurious features do not exist, posing a great challenge to robust model generalization.

Mitigating spurious bias typically depends on accurate annotations of spurious correlations between spurious features and classes, termed *group labels*. A group label (class, spurious feature) annotates a sample with a spurious feature in addition to its class label, providing a more granular categorization of data. For example, the Waterbirds dataset shown in Fig. 1 can be divided into four groups: (landbird, land), (landbird, water), (waterbird, land), and (waterbird, water). Models with spurious bias typically perform well on the majority groups which contain the majority of data, i.e., (landbird, land) and (waterbird, water), and perform poorly on the other groups, e.g., (landbird, water) and (waterbird, land), where the spurious correlations are different from those in the majority groups. Group labels play an important role in spurious bias mitigation, enabling direct performance optimization (Sagawa et al., 2019; Deng et al., 2024) and model selection (Liu et al., 2021; Kirichenko et al., 2023) under

known spurious correlations. However, group labels often require costly human-guided annotations, which are hard to acquire.

Removing the dependency on group labels allows us to tackle spurious bias in practically any scenarios where ERM training is adopted. However, this also opens up new challenges for **unsupervised spurious bias mitigation** where robustness to spurious correlations is not specified a priori by group labels. Recently, last-layer retraining (Kirichenko et al., 2023; Izmailov et al., 2022; LaBonte et al., 2024), which adjusts the weights of the last classification layer of an ERM model, has been successful in spurious bias mitigation guided by a held-out retraining set with group labels. The success demonstrates that neurons in the penultimate layer (before the last layer) provide sufficient information to tackle the prediction task at hand, as long as their contributions to final predictions are properly adjusted. This motivates us to detect neurons that are affected by spurious bias in order to mitigate it in the model. Although some existing methods (Singla & Feizi, 2021; Neuhaus et al., 2022) exploit neuron activations to detect spurious features, they require a certain amount of human supervision. The challenge that we aim to tackle is: *can we identify neurons*



Figure 1: The Waterbirds dataset (Sagawa et al., 2019). Training samples are partitioned into four groups: (landbird, land), (landbird, water), (waterbird, land), and (waterbird, water).

*affected by spurious bias without external supervision, e.g., group labels, and mitigate spurious bias accordingly?*

In this paper, for the first time, we theoretically demonstrate that neuron activations before the last classification layer, coupled with their final prediction outcomes, provide self-identifying information on whether the neurons are affected by spurious bias. Central to our theory is a term in a neuron activation that contributes to a model's spurious prediction behavior, which algins with the empirical observation that if representative samples with high activations on a neuron (Bykov et al., 2023; Singla & Feizi, 2021) are misclassified, then the neuron tends to be affected by spurious bias. Leveraging this insight, we propose a novel self-guided neuron detection method that works right before the last prediction layer to identify what neurons are affected by spurious bias for the given prediction task. With the incorporation of this method, we propose a last-layer selective activation retraining (LaSAR) framework that aims to retrain the last layer for improved robustness to spurious bias. During retraining, LaSAR is aware of the spuriousness of input neurons to the last prediction layer and selectively blocks the signals from the affected neurons. In this way, we promote the model to discover robust decision rules beyond spurious correlations.

We theoretically prove that LaSAR can effectively identify neurons affected by spurious bias and bring a model closer to the unbiased one. Our method LaSAR works in a classic ERM training setting where no additional annotations beyond class labels are available, which makes it a practical and efficient post-hoc tool for mitigating the spurious bias in a model. LaSAR is fully unsupervised in the sense that it does not requires external supervision, such as group labels, to mitigate a model's spurious bias. The ability to detect neurons affected by spurious bias in the latent space allows our method to be applicable to various data modalities, including vision and text data. Experiments show that our method outperforms baseline approaches in mitigating spurious bias across four benchmark datasets.

## 2 RELATED WORK

Exploiting spurious correlations for predictions has been demonstrated to be harmful to a model's generalization (Nushi et al., 2018; Zhang et al., 2018b; Geirhos et al., 2019; Clark et al., 2019; Nauta et al., 2021; Geirhos et al., 2020; Xiao et al., 2021). Thus, it is critical to mitigate the reliance on spurious correlations, or spurious bias, in models. In the following, we summarize existing methods into *supervised, semi-supervised, and unsupervised spurious bias mitigation*, based on the degrees of availability of external supervision.

**Supervised spurious bias mitigation.** In this setting, certain spurious correlations in data are given in the form of group labels. Spurious bias in a model is often demonstrated when there is a large gap between the model's average performance and its worst-group performance, indicating a strong reliance on certain spurious correlations that are not shared across groups of data. With group labels in the training data, balancing the size of the groups (Cui et al., 2019; He & Garcia, 2009), upweighting groups that do not have specified spurious correlations (Byrd & Lipton, 2019), or optimizing the worst-group performance (Sagawa et al., 2019) can be effective. Regularization strategies, such as using information bottleneck (Tartaglione et al., 2021) or the distributional distance between bias-aligned samples (Barbano et al., 2023), are also proved to be effective in spurious bias mitigation. A recent work (Wang et al., 2024) exploits the concept of neural collapse for spurious bias mitigation. However, this setting requires to know what spurious bias needs to be mitigated a priori and only focuses on mitigating the specified spurious bias.

**Semi-supervised spurious bias mitigation.** This setting relaxes the requirement of group labels in the training data but does require a small portion in a held-out set for achieving optimal performance. In other words, the goal is to mitigate targeted spurious bias without extensive spurious correlation annotations. One line of works is to use data augmentation, such as mixup (Zhang et al., 2018a; Han et al., 2022; Wu et al., 2023) or selective augmentation (Yao et al., 2022), to mitigate spurious bias in model training. Additionally, some methods propose to infer group labels in the training data using misclassified samples (Liu et al., 2021), clustering hidden embeddings (Zhang et al., 2022), or training a group label estimator (Nam et al., 2022) with a part of group-annotated validation data. Creager et al. (2021) infers group labels and adopts invariant learning. Moreover, Bahng et al. (2020) uses biased models to represent certain spurious biases, Zhang et al. (2024) improves bias learning and mitigation via poisoning attack, and Zhang et al. (2023) exploits the training dynamics to mine intermediate attribute samples for spurious bias mitigation. Last layer retraining (Kirichenko et al., 2023) uses a half of group-balanced validation data to retrain the last layer of a model. Recently, LaBonte et al. (2024) relaxes the requirement of group labels in one-half of the validation data using the early-stop disagreement criterion for selecting retraining samples. We also adopt last layer retraining but focus on a completely different setting where no group labels are available for training.

**Unsupervised spurious bias mitigation.** This setting does not assume any knowledge about spurious correlations in data, and the goal is to train a robust model that works well on certain data with known spurious correlations. Typically, we would expect relatively lower performance for methods working in this setting than in the other two settings as no information regarding the spurious correlations in test data is provided. A recent method (Li et al., 2024) upweights the training samples that are misclassified by a bias-amplified model and selects models using minimum class difference. Our method also works in this challenging setting. We take inspiration from spurious feature detection using neuron activations (Singla & Feizi, 2022; Neuhaus et al., 2022) but fully automate this process and integrate into our spurious bias mitigation framework. We propose a novel spuriousness fitness score to select robust models.

## 3 METHODOLOGY

### 3.1 PROBLEM SETTING

We consider a standard classification problem in which we assume that the dataset $\mathcal{D}_{\text{train}} = \{(\mathbf{x}, y) | \mathbf{x} \in \mathcal{X}, y \in \mathcal{Y}\}$ can be partitioned into groups $\mathcal{D}_g^{\text{tr}}$ with $\mathcal{D}_{\text{train}} = \cup_{g \in \mathcal{G}} \mathcal{D}_g^{\text{tr}}$, where $\mathbf{x}$ denotes a sample in the input space $\mathcal{X}$, $y$ is the corresponding label in the finite label space $\mathcal{Y}$, $g := (y, a)$ denotes the group label defined by the combination of a class label $y$ and a spurious feature $a \in \mathcal{A}$, where $\mathcal{A}$ denotes all spurious features in $\mathcal{D}_{\text{train}}$, and $\mathcal{G}$ denotes all possible group labels. A group of sample-label pairs in $\mathcal{D}_g^{\text{tr}}$ have the same class label $y$ and the same spurious feature $a$.

**Our scenario: unsupervised spurious bias mitigation.** In this setting, no group labels are available, resembling the traditional ERM training. In this setting, it is challenging to train a model $f_{\boldsymbol{\theta}}$ that is *robust to unknown spurious correlations* in the given dataset $\mathcal{D}_{\text{train}}$. A commonly used performance measure is the worst-group accuracy (WGA), which is the accuracy on the worst performing data group in the test set $\mathcal{D}_{\text{test}}$, i.e., $\text{WGA} = \min_{g \in \mathcal{G}} \text{Acc}(f_{\boldsymbol{\theta}}, \mathcal{D}_g^{\text{te}})$, where $\mathcal{D}_g^{\text{te}}$ denotes a group of data in $\mathcal{D}_{\text{test}}$ with $\mathcal{D}_{\text{test}} = \cup_{g \in \mathcal{G}} \mathcal{D}_g^{\text{te}}$. Typically, data in $\mathcal{D}_{\text{train}}$ is unbalanced across groups, and the trained

Figure 2: Method overview. (a) Extract latent embeddings (neuron activations) and prediction outcomes from an ERM-trained model using the identification data $\mathcal{D}_{\text{Ide}}$. (b) Identify dimensions (neurons) affected by spurious bias utilizing prediction outcomes (red for correct and blue for incorrect predictions). (c) Retrain the last prediction layer using selective activations on $\mathcal{D}_{\text{Ret}}$.

model $f_{\boldsymbol{\theta}}$ tends to favor certain data groups and to have a low WGA. Improving WGA without the guidance of group labels is challenging.

To tackle this, we first propose a practical and efficient retraining framework (Section 3.2) for spurious bias mitigation, which utilizes the self-identifying information of spurious bias contained in neuron activations along with their final prediction outcomes. Next, we provide a theoretical analysis (Section 3.3) to justify our design choices.

## 3.2 LAST LAYER SELECTIVE ACTIVATION RETRAINING

### 3.2.1 IDENTIFYING AFFECTED NEURONS

We first focus on identifying dimensions (neurons) from latent embeddings (neuron activations) of a targeted model that are affected by spurious bias. Identifying affected neurons allows us to design a general detection method independent of the data modality adopted in training.

Consider that we are given a well-trained ERM model $f_{\boldsymbol{\theta}}$ with parameters $\boldsymbol{\theta}$ as follows

$$\boldsymbol{\theta} = \arg\min_{\boldsymbol{\theta}'} \mathbb{E}_{(\mathbf{x},y)\in\mathcal{D}_{\text{train}}} \ell(f_{\boldsymbol{\theta}'}(\mathbf{x}), y), \tag{1}$$

where $\ell$ denotes the cross-entropy loss function. The model $f_{\boldsymbol{\theta}} = e_{\boldsymbol{\theta}_1} \circ h_{\boldsymbol{\theta}_2}$ consists of a feature extractor $e_{\boldsymbol{\theta}_1} : \mathcal{X} \to \mathbb{R}^M$ followed by a classifier $h_{\boldsymbol{\theta}_2} : \mathbb{R}^M \to \mathbb{R}^{|\mathcal{Y}|}$, where $M$ is the number of dimensions of latent embeddings obtained after $e_{\boldsymbol{\theta}_1}$, $\circ$ denotes the function composition operator, and $\boldsymbol{\theta} = \boldsymbol{\theta}_1 \cup \boldsymbol{\theta}_2$. Here, $h_{\boldsymbol{\theta}_2}$ is the last linear layer of the model with parameters $\boldsymbol{\theta}_2$, and $e_{\boldsymbol{\theta}_1}$ represents the remaining layers. As shown in Fig. 2(a), we extract a set of latent embeddings and prediction outcomes from the identification data $\mathcal{D}_{\text{Ide}}$ for the class $y$, i.e.,

$$\mathcal{V}^y = \{(\mathbf{v}_n, o_n) | \mathbf{v}_n = e_{\boldsymbol{\theta}_1}(\mathbf{x}_n), o_n = \mathbb{1}\{\arg\max f_{\boldsymbol{\theta}}(\mathbf{x}_n) == y_n\}, (\mathbf{x}_n, y_n) \in \mathcal{D}_{\text{Ide}}\}, \tag{2}$$

where $\mathbf{v}_n \in \mathbb{R}^M$ is an $M$-dimensional latent embedding for $\mathbf{x}_n$, and $o_n$ is the corresponding prediction outcome with $\mathbb{1}$ being an indicator function. We use the held-out validation data $\mathcal{D}_{\text{val}}$ as the identification data.

With the set of latent embeddings and prediction outcomes $\mathcal{V}^y$, we first propose a novel score termed *spuriousness score* $\delta_i^y$, which measures the spuriousness of the $i$'th dimension for predicting the class $y$. A larger spuriousness score indicates that the corresponding dimension is more likely to be affected by the spurious bias in the model. To calculate $\delta_i^y$, we first group $\mathcal{V}^y$ at the $i$'th dimension into correctly and incorrectly predicted sets $\hat{\mathcal{V}}_i^y$ and $\bar{\mathcal{V}}_i^y$, respectively:

$$\hat{\mathcal{V}}_i^y = \{\mathbf{v}_n[i] | (\mathbf{v}_n, o_n) \in \mathcal{V}^y, o_n = 1\}, \quad \forall i = 1, \ldots, M, \ y \in \mathcal{Y}, \tag{3}$$

and

$$\bar{\mathcal{V}}_i^y = \{\mathbf{v}_n[i] | (\mathbf{v}_n, o_n) \in \mathcal{V}^y, o_n = 0\}, \quad \forall i = 1, \ldots, M, \ y \in \mathcal{Y}, \tag{4}$$

where $\mathbf{v}_n[i]$ denotes the $i$'th element in $\mathbf{v}_n$. As illustrated in Fig. 2(b), we define $\delta_i^y$ as follows:

$$\delta_i^y = \mu_{\text{mis}} - \mu_{\text{cor}} = \text{Med}(\bar{\mathcal{V}}_i^y) - \text{Med}(\hat{\mathcal{V}}_i^y), \tag{5}$$

where $\text{Med}(\cdot)$ gets the median from a set of values. A high $\mu_{\text{mis}}$ indicates that high activations at the $i$'th dimension has adverse effects on predicting the class $y$, while a low $\mu_{\text{cor}}$ implies that low activations at the $i$'th dimension has little effect on the predictions. Thus, a large difference between $\mu_{\text{mis}}$ and $\mu_{\text{cor}}$, i.e., a large $\delta_i^y$, indicates a high likelihood of the $i$'th dimension being affected by the spurious bias in the model, i.e., the model incorrectly amplifies a spurious feature in the neuron activation when it should not. In contrast, a negative $\delta_i^y$ shows the importance of the $i$'th dimension for predictions as most correctly predicted samples tend to have high activation values on this dimension, while most incorrectly predicted samples have low activation values. Therefore, we set a cutoff value of 0 to select dimensions affected by spurious bias as follows:

$$\mathcal{S} = \{i | \delta_i^y > 0, \forall i = 1, \ldots, M, y \in \mathcal{Y}\}. \tag{6}$$

While our approach may resemble traditional variable selection, it goes further by specifically addressing spurious bias—a factor often ignored in traditional methods. Additionally, it operates in an unsupervised setting without requiring group annotations. Further details on its advantages are provided in Appendix.

Note that an identified dimension for one class cannot serve as a key contributor to predicting some other class. For example, when the goal is to classify between a "rectangle" and the "blue color", the dimension with a strong reliance on the "blue color" for the "rectangle" class cannot be used to predict the "blue color" class given a blue rectangle, as the prediction will be ambiguous. Therefore, $\mathcal{S}$ includes the identified dimensions for all the classes.

In the following, we refer to a dimension as a **spurious dimension** when $\delta_i^y > 0$ and a **core dimension** when $\delta_i^y < 0$. However, these terms do not imply that a dimension exclusively represents either spurious or core features. In practice, a core dimension exhibits high activation values for the target class, whereas a spurious dimension shows high activation values for an undesired class. Visualizations of several identified spurious and core dimensions on real-world datasets are provided in Figs. 5 through 8 in Appendix.

### 3.2.2 MITIGATE SPURIOUS BIAS

**Learning objective.** With the identified spurious dimensions, we propose to selectively retrain the last prediction layer to mitigate the reliance on spurious correlations. As illustrated in Fig. 2(c), during retraining, we selectively activate dimensions (neurons) that are not identified as spurious while masking out the signals from spurious dimensions. In this way, we explicitly break the correlations between spurious features and prediction targets and promote the model to discover robust decision rules beyond spurious correlations. Concretely, given a retraining dataset $\mathcal{D}_{\text{Ret}}$, we optimize the last classification layer as follows,

$$\boldsymbol{\theta}_2^* = \arg\min_{\boldsymbol{\theta}_2} \mathbb{E}_{\mathcal{B} \sim \mathcal{D}_{\text{Ret}}} \ell(h_{\boldsymbol{\theta}_2}(\tilde{\mathbf{v}}_n), y), \tag{7}$$

where $\mathcal{B}$ is a batch containing *class-balanced* sample-label pairs from $\mathcal{D}_{\text{Ret}}$, avoiding the classifier favoring certain classes during retraining, and $\tilde{\mathbf{v}}_n$ is the latent embedding after zeroing-out activations on the identified spurious dimensions $\mathcal{S}$. Unless otherwise stated, we use $\mathcal{D}_{\text{train}}$ as $\mathcal{D}_{\text{Ret}}$.

**Model selection.** Without group labels, we have no knowledge about what spurious correlations a model might capture during training, which is challenging to select robust models (Liu et al., 2021; Yang et al., 2023). We address this by designing a novel model selection metric, termed *spuriousness fitness score (SFit)*, based on our proposed spuriousness score. We calculate SFit as follows:

$$\text{SFit} = \sum_{m=1}^{M} \sum_{y \in \mathcal{Y}} \text{Abs}(\delta_m^y), \tag{8}$$

where $\text{Abs}(\cdot)$ returns the absolute value of a given input. In practice, a high SFit can select a robust model that has easily self-distinguishable spurious and core dimensions.

We use Equation (6) and Equation (7) to perform spurious dimension detection and spurious bias mitigation iteratively and use SFit for model selection. Our method, termed *last layer selective activation retraining* (LaSAR), works in the unsupervised spurious bias mitigation setting and is very efficient in retraining as only the last layer is involved.

### 3.3 THEORETICAL ANALYSIS

#### 3.3.1 PRELIMINARY

We consider the following setting which is feasible for a theoretical analysis while capturing the essence of our proposed method, LaSAR. We first model a sample-label pair $(\mathbf{x}, y)$ following the standard setting in Arjovsky et al. (2019); Ye et al. (2023):

$$\mathbf{x} = (\mathbf{x}_{\text{core}}, \mathbf{x}_{\text{spu}})^T \in \mathbb{R}^{D \times 1}, \; y = \boldsymbol{\beta}^T \mathbf{x}_{\text{core}} + \varepsilon_{\text{core}}, \tag{9}$$

where the core component $\mathbf{x}_{\text{core}} \in \mathbb{R}^{D_1 \times 1}$ follows some distribution $\mathbb{P}$, and the spurious component $\mathbf{x}_{\text{spu}} \in \mathbb{R}^{D_2 \times 1}$ with $D_1 + D_2 = D$ is associated with the label $y$ with the following relation:

$$\mathbf{x}_{\text{spu}} = (2a - 1)\boldsymbol{\gamma} y + \boldsymbol{\varepsilon}_{\text{spu}}, a \sim \text{Bern}(p), \tag{10}$$

where $(2a - 1) \in \{-1, +1\}$, $a \sim \text{Bern}(p)$ is a Bernoulli random variable, and $p$ is close to 1, indicating that $\mathbf{x}_{\text{spu}}$ is mostly indicative of $y$ but not always. In Equation (9) and Equation (10), $\boldsymbol{\beta} \in \mathbb{R}^{D_1 \times 1}$ and $\boldsymbol{\gamma} \in \mathbb{R}^{D_2 \times 1}$ are coefficients with unit $\ell_2$ norm, and $\varepsilon_{\text{core}}$ and $\boldsymbol{\varepsilon}_{\text{spu}}$ model the variations in the core and spurious components, respectively. We set $\varepsilon_{\text{core}}$ and each element in $\boldsymbol{\varepsilon}_{\text{spu}}$ as a zero-mean Gaussian random variable with the variance $\eta_{\text{core}}^2$ and $\eta_{\text{spu}}^2$, respectively. We set $\eta_{\text{core}}^2 \gg \eta_{\text{spu}}^2$ to facilitate the learning of spurious features (Sagawa et al., 2019).

To capture the property of latent features, we consider a regression task using a commonly adopted two-layer linear network (Ye et al., 2023) defined as $f(\mathbf{x}) = \mathbf{b}^T \mathbf{W}\mathbf{x}$, where $\mathbf{W} \in \mathbb{R}^{M \times D}$ denotes the embedding function, and $\mathbf{b} \in \mathbb{R}^{M \times 1}$ denotes the last layer. The model $f(\mathbf{x})$ can be further expressed as follows,

$$f(\mathbf{x}) = \sum_{i=1}^{M} b_i(\mathbf{x}_{\text{core}}^T \mathbf{w}_{\text{core},i} + \mathbf{x}_{\text{spu}}^T \mathbf{w}_{\text{spu},i}) = \mathbf{x}_{\text{core}}^T \mathbf{u}_{\text{core}} + \mathbf{x}_{\text{spu}}^T \mathbf{u}_{\text{spu}}, \tag{11}$$

where $\mathbf{w}_i^T \in \mathbb{R}^{1 \times D}$ is the $i$'th row of $\mathbf{W}$, $\mathbf{w}_i^T = [\mathbf{w}_{\text{core},i}^T, \mathbf{w}_{\text{spu},i}^T]$ with $\mathbf{w}_{\text{core},i} \in \mathbb{R}^{D_1 \times 1}$ and $\mathbf{w}_{\text{spu},i} \in \mathbb{R}^{D_2 \times 1}$, $\mathbf{u}_{\text{core}} = \sum_{i=1}^{M} b_i \mathbf{w}_{\text{core},i}$, and $\mathbf{u}_{\text{spu}} = \sum_{i=1}^{M} b_i \mathbf{w}_{\text{spu},i}$. During the training stage, we minimize $\ell_{\text{tr}}(\mathbf{W}, \mathbf{b}) = \frac{1}{2}\mathbb{E}_{(\mathbf{x},y) \in \mathcal{D}_{\text{train}}} \|f(\mathbf{x}) - y\|_2^2$.

#### 3.3.2 MAIN RESULTS

**Proposition 1 (Principal for selective activation).** *Given the model $f(\mathbf{x}) = \mathbf{b}^T \mathbf{W}\mathbf{x}$ trained with data specified in Equation (9) and Equation (10), it captures spurious correlations when $\boldsymbol{\gamma}^T \mathbf{w}_{spu,i} < 0, i \in \{1, \dots, M\}$. The principal of selective activation is to mask out neurons containing negative $\boldsymbol{\gamma}^T \mathbf{w}_{spu,i}$. The proof is in Appendix.*

**Remark.** If $\boldsymbol{\gamma}^T \mathbf{w}_{\text{spu},i} \geq 0$, the model handles the spurious component correctly. Specifically, when $a = 1$, the spurious component $\mathbf{x}_{\text{spu}}$ positively correlates with the core component $\mathbf{x}_{\text{core}}$ and contributes to the output, whereas when $a = 0$, its correlation with $\mathbf{x}_{\text{core}}$ breaks with a negative one and has a negative contribution to the output. The relations reverse when $\boldsymbol{\gamma}^T \mathbf{w}_{\text{spu},i} < 0$, i.e., the model still utilizes $\mathbf{x}_{\text{spu}}$ even when the correlation breaks, demonstrating a strong reliance on the spurious component instead of the core component.

**Lemma 1.** *Given a training dataset $\mathcal{D}_{train}$ with $p$ defined in Equation (10) satisfying $1 \geq p \gg 0.5$, the optimized weights in the form of $\mathbf{u}_{core}^*$ and $\mathbf{u}_{spu}^*$ are*

$$\mathbf{u}_{core}^* = \frac{(2 - 2p)\eta_{core}^2 + \eta_{spu}^2}{\eta_{core}^2 + \eta_{spu}^2}\boldsymbol{\beta}, \; \mathbf{u}_{spu}^* = \frac{(2p - 1)\eta_{core}^2}{\eta_{core}^2 + \eta_{spu}^2}\boldsymbol{\gamma}. \tag{12}$$

**Remark.** When $p = 0.5$, the training data is unbiased and we obtain an unbiased classifier with weights $\mathbf{u}_{\text{core}}^* = \boldsymbol{\beta}$ and $\mathbf{u}_{\text{spu}}^* = 0$. The proof is in Appendix.

**Theorem 1 (Metric for neuron selection).** *Given the model $f(\mathbf{x}) = \mathbf{b}^T \mathbf{W}\mathbf{x}$, we cast it to a classification model by training it to regress $y \in \{-\mu, \mu\}$ ($\mu > 0$) on $\mathbf{x}$ based on the data model specified in Equation (9) and Equation (10), where $\mu = \mathbb{E}[\boldsymbol{\beta}^T \mathbf{x}_{core}]$. The metric $\delta_i^y$ defined in the following can identify neurons with spurious correlations when $\delta_i^y > 0$:*

$$\delta_i^y = Med(\bar{\mathcal{V}}_i^y) - Med(\hat{\mathcal{V}}_i^y),$$

*where $\bar{\mathcal{V}}_i^y$ and $\hat{\mathcal{V}}_i^y$ are the sets of activation values for misclassified and correctly predicted samples with the label $y$ from the $i$'th neuron, respectively; $Med(\cdot)$ denotes the Median operator; and an activation value is defined as $\mathbf{x}_{core}^T \mathbf{w}_{core,i} + \mathbf{x}_{spu}^T \mathbf{w}_{spu,i}$. The proof is in Appendix.*

**Remark.** The theorem establishes that $\delta_i^y \approx -2\mu\gamma^T \mathbf{w}_{\text{spu},i}$, which proves that our neuron selection metric defined in Equation (6) follows the principal in Proposition 1 and can select spurious dimensions.

**Theorem 2 (LaSAR mitigates spurious bias).** *Consider the model $f^*(\mathbf{x}) = \mathbf{x}^T \mathbf{u}^*$ trained on the biased training data with $p \gg 0.5$, with $\mathbf{u}_{core}^*$ and $\mathbf{u}_{spu}^*$ defined in Equation (12). Under the mild assumption that $\boldsymbol{\beta}^T \mathbf{w}_{core,i} \approx \boldsymbol{\gamma}^T \mathbf{w}_{spu,i}, \forall i = 1, \ldots, M$, then applying LaSAR to $f^*(\mathbf{x})$ produces a model that is closer to the unbiased one. The proof is in Appendix.*

**Remark.** The assumption that $\boldsymbol{\beta}^T \mathbf{w}_{\text{core},i} \approx \boldsymbol{\gamma}^T \mathbf{w}_{\text{spu},i}, \forall i = 1, \ldots, M$ generally holds for a biased model as the model has learned to associate spurious features with the core features. Denote the LaSAR solutions as $\mathbf{u}_{\text{core}} = \mathbf{u}_{\text{core}}^{\dagger}$ and $\mathbf{u}_{\text{spu}} = \mathbf{u}_{\text{spu}}^{\dagger}$. An interesting finding is that retraining the last layer does not alter the weight on the spurious component, i.e., $\mathbf{u}_{\text{spu}}^{\dagger} = \mathbf{u}_{\text{spu}}^*$, which is the optimal solution achievable by last-layer retraining methods (see Lemma 3 in Appendix). However, it does adjust $\mathbf{u}_{\text{core}}^{\dagger}$ to be closer to the optimal weight on the core component, $\boldsymbol{\beta}$. Overall, LaSAR brings model parameters closer to the optimal, unbiased solution compared to the parameters of the original biased model. Moreover, unlike sample-level last-layer retraining methods, such as AFR (Qiu et al., 2023), LaSAR is guaranteed to outperform the ERM-trained model. Additional discussions on this topic can be found in Appendix.

## 4 EXPERIMENT

### 4.1 DATASETS

We tested LaSAR on four image datasets and two text datasets with various types of spurious features: (1) **Waterbirds** (Sagawa et al., 2019) is an image dataset for recognizing waterbirds and landbirds. It is generated synthetically by combining images of the two kinds of birds from the CUB dataset (Welinder et al., 2010) and the backgrounds, water and land, from the Places dataset (Zhou et al., 2017). (2) **CelebA** (Liu et al., 2015) is a large-scale image dataset of celebrity faces. The task is to identify hair color, non-blond or blond, with male and female as the spurious features. (3) **ImageNet-9** Xiao et al. (2021) is a subset of ImageNet Deng et al. (2009) containing nine super-classes. It comprises images with different background and foreground signals and can be used to assess how much models rely on image backgrounds. (4) **ImageNet-A** Hendrycks et al. (2021) is a dataset of real-world images, adversarially curated to test the limits of classifiers such as ResNet-50. We use this dataset to test the robustness of a classifier after training it on ImageNet-9. (5) **MultiNLI** (Williams et al., 2017) is a text classification dataset with 3 classes: neutral, contradiction, and entailment, representing the natural language inference relationship between a premise and a hypothesis. The spurious feature is the presence of negation, which is highly correlated with the contradiction label. Standard train/validation/test splits are used as provided by prior work. (6) **CivilComments** (Borkan et al., 2019) is a binary classification text dataset aimed at predicting whether an internet comment contains toxic language. The spurious feature involves references to eight demographic identities: male, female, LGBTQ, Christian, Muslim, other religions, Black, and White. The dataset uses standard splits provided by the WILDS benchmark (Koh et al., 2021).

### 4.2 EXPERIMENTAL SETUP

**Training details.** We first train ERM models on each of the four datasets. For image datasets, we use ResNet-50 and ResNet-18 models (He et al., 2016) pretrained on ImageNet, while for text datasets, we use a BERT model (Kenton & Toutanova, 2019) pretrained on Book Corpus and English Wikipedia data. We follow the settings in (Izmailov et al., 2022) for ERM training. The best ERM models are selected based on the average validation accuracy. For our LaSAR training, we first identify spurious dimensions using $\mathcal{D}_{\text{Ide}}$ and retrain a given ERM model using $\mathcal{D}_{\text{Ret}}$. For nonnegative neuron activations, we take their absolution values before the identification process. We run the training

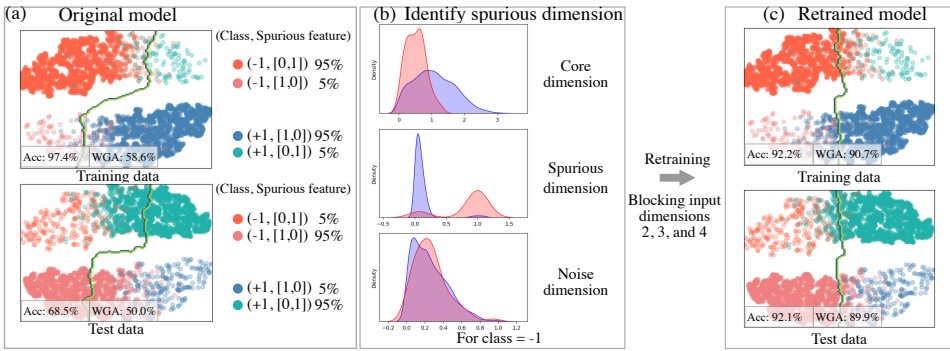

Figure 3: Illustration of our motivating example. (a) Visualization of training and test data using t-SNE (van der Maaten & Hinton, 2008) along with the decision boundaries of the trained model. (b) Identify spurious dimensions for $y = -1$ based on the discrepancy of value distributions for the correctly (blue) and incorrectly (red) predicted samples. (c) Retraining the model while blocking identified input dimensions improves WGA. The figure is best viewed in color.

under three different random seeds and report average accuracies along with standard deviations. We ran all experiments on NVIDIA RTX 8000 GPUs. We report full training details in Appendix.

**Evaluation metrics.** To evaluate the robustness to spurious bias, we adopt the widely accepted robustness metric, *worst-group accuracy (WGA)*, that gives the lower-bound performance of a classifier on the test set with various dataset biases. We also focus on the *accuracy gap* between the standard average accuracy and the worst-group accuracy as a measure of a classifier's reliance on spurious correlations. A high worst-group accuracy and a low accuracy gap indicate that the classifier is robust to spurious correlations and can fairly predict samples from different groups.

### 4.3 SYNTHETIC EXPERIMENT

**Preliminary.** Without loss of generality, we consider an input $\mathbf{v} \in \mathbb{R}^4$ to simulate a latent embedding before the last prediction layer. This embedding consists of three components: a core feature $v^c \in \mathbb{R}$, a spurious feature $\mathbf{v}^s \in \mathbb{R}^2$, and a noise feature $v^\epsilon \in \mathbb{R}$. We generate a synthetic training dataset with labels $\{-1, +1\}$, where the core features are perturbed version of the labels. The spurious feature $\mathbf{u}^s$ is generated such that for 95% of the samples with $y_i = -1$, it is a perturbed version of $[0, 1]$, while for the remaining 5%, it is a perturbed version of $[1, 0]$. The other cases are similarly illustrated in Fig. 3(a). As $\mathbf{v}$ represents a latent embedding, we thus consider a logistic regression model $\phi_{\tilde{\mathbf{w}}}(\mathbf{v}) = 1/(1 + \exp\{-(\mathbf{w}^T \mathbf{v} + b)\})$, where $\tilde{\mathbf{w}} = [\mathbf{w}, b]$. The model predicts $+1$ when $\phi_{\tilde{\mathbf{w}}}(\mathbf{v}) > 0.5$ and $-1$ otherwise. We trained $\phi_{\tilde{\mathbf{w}}}$ on the synthetic training data and tested it on the corresponding test data. Further details are provided in the Appendix.

**Results.** The top plot in Fig. 3(b) shows the distribution of the first dimension of input embeddings when $y_i = -1$. In contrast, for the noise dimension (i.e., the fourth dimension of $\mathbf{v}$), randomness results in negligible differences between the two distributions, as illustrated in the bottom plot of Fig. 3(b). Additional plots for all dimensions can be found in Fig. 4 in the Appendix. We then retrained the model while blocking the second, third, and fourth dimensions. The retrained model has learned to better balance its performance on both the training and test data, resulting in a substantial improvement in WGA on the test data (Fig. 3(c)). Importantly, this process relies solely on the intrinsic characteristics of the model and does not require external supervisions.

### 4.4 COMPARISON WITH EXISTING METHODS

We evaluated our method against existing approaches designed to address spurious bias on both image and text datasets. Our primary focus was on methods specifically developed for unsupervised spurious bias mitigation, where no group labels are available to guide the mitigation process. To provide additional context, we also

| Method | Waterbirds | CelebA |
|--------|-----------|--------|
| JTT | $84.2_{\pm 0.5}$ | $52.3_{\pm 1.8}$ |
| AFR | $89.0_{\pm 2.6}$ | $68.7_{\pm 1.7}$ |
| LaSAR | $91.8_{\pm 0.8}$ | $83.0_{\pm 2.8}$ |

Table 1: WGA comparison.

| Algorithm | Group annotations | | Waterbirds | | | CelebA | | |
|---|---|---|---|---|---|---|---|---|
| | Train | Val | WGA ($\uparrow$) | Acc. ($\uparrow$) | Acc. Gap ($\downarrow$) | WGA ($\uparrow$) | Acc. ($\uparrow$) | Acc. Gap ($\downarrow$) |
| JTT (Liu et al., 2021) | No | Yes | 86.7 | 93.3 | 6.6 | 81.1 | 88.0 | 6.9 |
| SELF[†] (LaBonte et al., 2024) | No | Yes | $93.0_{\pm 0.3}$ | $94.0_{\pm 1.7}$ | **1.0** | $83.9_{\pm 0.9}$ | $91.7_{\pm 0.4}$ | 7.8 |
| CNC (Zhang et al., 2022) | No | Yes | $88.5_{\pm 0.3}$ | $90.9_{\pm 0.1}$ | 2.4 | $88.8_{\pm 0.9}$ | $89.9_{\pm 0.5}$ | **1.1** |
| BAM (Li et al., 2024) | No | Yes | $89.2_{\pm 0.3}$ | $91.4_{\pm 0.4}$ | 2.2 | $83.5_{\pm 0.9}$ | $88.0_{\pm 0.4}$ | 4.5 |
| AFR[†] (Qiu et al., 2023) | No | Yes | $90.4_{\pm 1.1}$ | $94.2_{\pm 1.2}$ | 3.8 | $82.0_{\pm 0.5}$ | $91.3_{\pm 0.3}$ | 9.3 |
| DFR[†] (Kirichenko et al., 2023) | No | Yes | $92.4_{\pm 0.9}$ | $94.9_{\pm 0.3}$ | 2.5 | $87.0_{\pm 1.1}$ | $92.6_{\pm 0.5}$ | 5.6 |
| ERM (Vapnik, 1999) | No | No | 72.6 | **97.3** | 24.7 | 47.2 | **95.6** | 48.4 |
| BPA (Seo et al., 2022) | No | No | 71.4 | - | - | 82.5 | - | - |
| GEORGE (Sohoni et al., 2020) | No | No | 76.2 | 95.7 | 19.5 | 52.4 | 94.8 | 42.4 |
| BAM (Li et al., 2024) | No | No | $89.1_{\pm 0.2}$ | $91.4_{\pm 0.3}$ | 2.3 | $80.1_{\pm 3.3}$ | $88.4_{\pm 2.3}$ | 8.3 |
| LaSAR | No | No | $91.8_{\pm 0.8}$ | $94.0_{\pm 0.2}$ | **2.2** | $83.0_{\pm 2.8}$ | $92.0_{\pm 0.5}$ | 9.0 |
| LaSAR[†] | No | No | $91.7_{\pm 1.2}$ | $94.4_{\pm 0.4}$ | 2.7 | $87.4_{\pm 0.4}$ | $90.3_{\pm 0.7}$ | **2.9** |

Table 2: Comparison of worst-group accuracy (%), average accuracy (%), and accuracy gap (%) on the image datasets. [†] denotes using a fraction of validation data for retraining.

| Algorithm | Group annotations | | MultiNLI | | | CivilComments | | |
|---|---|---|---|---|---|---|---|---|
| | Train | Val | WGA ($\uparrow$) | Acc. ($\uparrow$) | Acc. Gap ($\downarrow$) | WGA ($\uparrow$) | Acc. ($\uparrow$) | Acc. Gap ($\downarrow$) |
| JTT (Liu et al., 2021) | No | Yes | 72.6 | 78.6 | **6.0** | 69.3 | **91.1** | 21.8 |
| SELF[†] (LaBonte et al., 2024) | No | Yes | $70.7_{\pm 2.5}$ | $81.2_{\pm 0.7}$ | 10.5 | $79.1_{\pm 2.1}$ | $87.7_{\pm 0.6}$ | 8.6 |
| CNC (Zhang et al., 2022) | No | Yes | - | - | - | $68.9_{\pm 2.1}$ | $81.7_{\pm 0.5}$ | 12.8 |
| BAM (Li et al., 2024) | No | Yes | $71.2_{\pm 1.6}$ | $79.6_{\pm 1.1}$ | 8.4 | $79.3_{\pm 2.7}$ | $88.3_{\pm 0.8}$ | 9.0 |
| AFR[†] (Qiu et al., 2023) | No | Yes | $73.4_{\pm 0.6}$ | $81.4_{\pm 0.2}$ | 8.0 | $68.7_{\pm 0.6}$ | $89.8_{\pm 0.6}$ | 21.1 |
| DFR[†] (Kirichenko et al., 2023) | No | Yes | $70.8_{\pm 0.8}$ | $81.7_{\pm 0.2}$ | 10.9 | $81.8_{\pm 1.6}$ | $87.5_{\pm 0.2}$ | **5.7** |
| ERM (Vapnik, 1999) | No | No | 67.9 | **82.4** | 14.5 | 57.4 | **92.6** | 35.2 |
| BAM (Li et al., 2024) | No | No | $70.8_{\pm 1.5}$ | $80.3_{\pm 1.0}$ | 9.5 | $79.3_{\pm 2.7}$ | $88.3_{\pm 0.8}$ | 9.0 |
| LaSAR | No | No | $70.6_{\pm 0.4}$ | $81.5_{\pm 0.7}$ | 10.9 | $82.4_{\pm 0.2}$ | $89.2_{\pm 0.1}$ | **6.8** |
| LaSAR[†] | No | No | $72.4_{\pm 0.3}$ | $80.2_{\pm 0.6}$ | **7.8** | $73.6_{\pm 0.5}$ | $85.4_{\pm 0.2}$ | 11.8 |

Table 3: Comparison of worst-group accuracy (%), average accuracy (%), and accuracy gap (%) on the text datasets. [†] denotes using a fraction of validation data for retraining.

included methods designed for semi-supervised spurious bias mitigation to highlight the performance gap between the two settings.

We first compared our approach against AFR (Qiu et al., 2023) and JTT (Liu et al., 2021) to demonstrate the challenges of the unsupervised setting for semi-supervised methods. These methods were tuned using worst-class accuracy (Yang et al., 2023) on the validation set instead of WGA. As shown in Table 1, our method exhibits larger performance gains over AFR and JTT compared to their results presented in the subsequent tables.

The results in the lower part of Table 2 correspond to the unsupervised spurious bias mitigation setting, where no group labels are available. Our method, LaSAR, achieves the highest worst-group accuracies and the smallest accuracy gaps, demonstrating its effectiveness in enhancing model robustness to spurious bias while balancing performance across different data groups. The upper part of Table 2 presents results from the semi-supervised spurious bias mitigation setting. Even in this setting, LaSAR remains competitive, thanks to its strong spurious bias mitigation capabilities. On the text datasets, LaSAR continues to perform effectively, achieving the best worst-group accuracies and the smallest accuracy gaps in the unsupervised spurious bias mitigation setting, as shown in Table 3.

We further evaluated LaSAR on the more challenging ImageNet-9 (Kim et al., 2022; Bahng et al., 2020) and ImageNet-A (Hendrycks et al., 2021) datasets. Our approach involved first training an ERM model from scratch using the training data of ImageNet-9 and then fine-tuning the last layer with LaSAR. As shown in Table 4, LaSAR demonstrates a significant advantage by achieving the best performance on the challenging ImageNet-A dataset, which is known for its natural adversarial examples. While this improvement comes with a slight trade-off in in-distribution performance on ImageNet-9, it highlights LaSAR's ability to enhance robustness to distribution shifts, making it particularly effective in out-of-distribution scenarios.

| Method | Group annotations | ImageNet-9 | | ImageNet-A |
|---|---|---|---|---|
| | | Validation($\uparrow$) | Unbiased($\uparrow$) | Test($\uparrow$) |
| StylisedIN (Geirhos et al., 2018) | Yes | $88.4_{\pm0.5}$ | $86.6_{\pm0.6}$ | $24.6_{\pm1.4}$ |
| LearnedMixin (Clark et al., 2019) | Yes | $64.1_{\pm4.0}$ | $62.7_{\pm3.1}$ | $15.0_{\pm1.6}$ |
| RUBi (Cadene et al., 2019) | Yes | $90.5_{\pm0.3}$ | $88.6_{\pm0.4}$ | $27.7_{\pm2.1}$ |
| ERM (Vapnik, 1999) | No | $90.8_{\pm0.6}$ | $88.8_{\pm0.6}$ | $24.9_{\pm1.1}$ |
| ReBias (Bahng et al., 2020) | No | $91.9_{\pm1.7}$ | $90.5_{\pm1.7}$ | $29.6_{\pm1.6}$ |
| LfF (Nam et al., 2020) | No | 86.0 | 85.0 | 24.6 |
| CaaM (Wang et al., 2021) | No | **95.7** | **95.2** | 32.8 |
| SSL+ERM (Kim et al., 2022) | No | $94.2_{\pm0.1}$ | $93.2_{\pm0.0}$ | $34.2_{\pm0.5}$ |
| LWBC (Kim et al., 2022) | No | $94.0_{\pm0.2}$ | $93.0_{\pm0.3}$ | $36.0_{\pm0.5}$ |
| **LaSAR** | No | $93.7_{\pm0.1}$ | $92.4_{\pm0.0}$ | $\mathbf{37.3}_{\pm0.5}$ |

Table 4: Validation, Unbiased, and Test metrics (%) evaluated on the ImageNet-9 and ImageNet-A datasets. All methods use ResNet-18 as the backbone. The best results are in **boldface**.

| $\mathcal{D}_{\text{Ide}}$ | $\mathcal{D}_{\text{Ret}}$ | SAR | Waterbirds | CelebA | MultiNLI | CivilComments |
|---|---|---|---|---|---|---|
| $\mathcal{D}_{\text{train}}$ | $\mathcal{D}_{\text{train}}$ | Yes | $78.0_{\pm2.3}$ | $58.5_{\pm1.2}$ | $42.0_{\pm10.5}$ | $80.0_{\pm10.5}$ |
| $\mathcal{D}_{\text{val}}$ | $\mathcal{D}_{\text{train}}$ | Yes | $\mathbf{91.8}_{\pm0.8}$ | $83.0_{\pm2.8}$ | $65.0_{\pm1.5}$ | $\mathbf{82.4}_{\pm0.2}$ |
| $\mathcal{D}_{\text{val}}$ | $\mathcal{D}_{\text{train}}$ | No | $82.7_{\pm0.4}$ | $53.9_{\pm0.0}$ | $63.4_{\pm0.7}$ | $81.5_{\pm0.5}$ |
| $\mathcal{D}_{\text{val}}/2$ | $\mathcal{D}_{\text{val}}/2$ | Yes | $91.7_{\pm1.2}$ | $\mathbf{87.4}_{\pm0.4}$ | $\mathbf{72.4}_{\pm0.3}$ | $73.6_{\pm0.5}$ |

Table 5: Comparison of worst-group accuracy (%) between different choices of $\mathcal{D}_{\text{Ide}}$ and $\mathcal{D}_{\text{Ret}}$ as well as the proposed selective activation retraining (SAR) on the four datasets.

## 4.5 ABLATION STUDY

We analyzed the effectiveness of our proposed components in Table 5. Specifically, we focused on different choices of the identification dataset $\mathcal{D}_{\text{Ide}}$ and the retraining dataset $\mathcal{D}_{\text{Ret}}$ as well as the effectiveness of using selective activation retraining (SAR) with identified spurious dimensions. When we used the training data to identify spurious dimensions, i.e., $\mathcal{D}_{\text{Ide}} = \mathcal{D}_{\text{train}}$, we observed a relatively low performance on each dataset. However, after switching to a held-out validation data $\mathcal{D}_{\text{val}}$, we observed significant performance improvement in comparison with the previous setting. This demonstrates the benefit of using a new and held-out dataset for discovering spurious dimensions and avoiding overfitting to a used dataset $\mathcal{D}_{\text{train}}$. By default, our method LaSAR uses $\mathcal{D}_{\text{val}}$ as $\mathcal{D}_{\text{Ide}}$. Next, we sought to analyze whether SAR is effective by disabling it during retraining, which effectively reduces LaSAR to class-balanced retraining. We observed consistent performance degradation across the four datasets, which validates the effectiveness of SAR across multiple datasets. Finally, inspired by the success of DFR (Kirichenko et al., 2023), which uses a half of the validation data for retraining, we divide $\mathcal{D}_{\text{val}}$ into two halves and use one half (denoted as $\mathcal{D}_{\text{val}}/2$) as $\mathcal{D}_{\text{Ide}}$ and the other half as $\mathcal{D}_{\text{Ret}}$. Different from DFR, our method does not use group labels in the validation data. We observed that this strategy can further boost the performance on the CelebA and MultiNLI datasets. We also observed a performance degradation on the CivilComments dataset, possibly arising from the imperfect splitting of $\mathcal{D}_{\text{val}}$. We leave this to our future work.

## 5 CONCLUSION

Mitigating spurious bias is critical to models' generalization. We considered a challenging yet realistic unsupervised spurious bias mitigation setting: mitigating spurious bias in models without group labels. We proposed a self-guided spurious bias mitigation framework by exploiting the distinct patterns in neuron activations (latent embeddings) right before the last prediction layer of a model. Our framework tackles spurious bias in two stages by first identifying spurious dimensions and then retraining the last prediction layer of the model using latent embeddings while blocking inputs from spurious dimensions. We theoretically validated our proposed approach and demonstrated the effectiveness of our spurious dimension identification by showing that these dimensions represent non-essential parts of input samples. Our method does not need additional training data and can be used on different data modalities and with different model architectures.

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

# A APPENDIX

The appendix is organized as follows:

## A.1 DETAILS FOR THE SYNTHETIC EXPERIMENT

**Data model.** Without loss of generality, we consider an input $\mathbf{v} \in \mathbb{R}^4$ to simulate a latent embedding before the last prediction layer, which consists of three components: a core feature $v^c \in \mathbb{R}$, a spurious feature $\mathbf{v}^s \in \mathbb{R}^2$, and a noise feature $v^\epsilon \in \mathbb{R}$. We generate a dataset $\mathcal{D}^{\text{syn}} = \{(\mathbf{v}_i, y_i)\}_{i=1}^N$ of $N$ sample-label pairs, where $y_i \in \{-1, +1\}$, $v_i^c = y_i + n_c$, and $v^\epsilon$ and $n_c$ are zero-mean Gaussian noises with variances $\sigma_\epsilon^2$ and $\sigma_c^2$, respectively. When $y_i = -1$, $\mathbf{v}_i^s = [0,1] + \mathbf{n}_s$ with the probability $\alpha$ and $\mathbf{v}_i^s = [1,0] + \mathbf{n}_s$ with the probability $1 - \alpha$; when $y_i = +1$, $\mathbf{v}_i^s = [1,0] + \mathbf{n}_s$ with the probability $\alpha$ and $\mathbf{v}_i^s = [0,1] + \mathbf{n}_s$ with the probability $1 - \alpha$, where $\mathbf{n}_s$ is a vector of two independent zero-mean Gaussian noises with the variance $\sigma_s^2$. We design a spurious feature as a two-dimensional vector so that each dimension uniquely represents a spurious pattern, i.e., occurrences of 1's and 0's controlled by $\alpha$, for each class. To reveal spurious bias, i.e., using the correlation between $\mathbf{u}_i^s$ and $y_i$ for predictions, we generate a training set $\mathcal{D}_{\text{train}}^{\text{syn}}$ with easy-to-learn spurious features by setting $\sigma_c^2 > \sigma_s^2$ and $\alpha \approx 1$ (Sagawa et al., 2020). Thus, the correlations between $\mathbf{v}_i^s$ and $y_i$ are predictive of $\alpha N$ expected labels. To demonstrate, we set $\sigma_c^2 = 0.5$, $\sigma_s^2 = 0.01$, $\sigma_\epsilon^2 = 0.1$, $\alpha = 0.95$, and $N = 5000$. We generate a test set $\mathcal{D}_{\text{test}}^{\text{syn}}$ with the same set of parameters except $\alpha = 0.1$. Now, spurious correlations between $\mathbf{v}_i^s$ and $y_i$ are only predictive of a small portion of the test samples. Fig. 3(a) shows four data groups along with their respective proportions in each class.

**Classification model.** As the input $\mathbf{v}$ is a latent embedding, we thus consider a logistic regression model $\phi_{\tilde{\mathbf{w}}}(\mathbf{v}) = 1/(1 + \exp\{-(\mathbf{w}^T \mathbf{v} + b)\})$, where $\tilde{\mathbf{w}} = [\mathbf{w}, b]$. The model predicts $+1$ when $\phi_{\tilde{\mathbf{w}}}(\mathbf{v}) > 0.5$ and $-1$ otherwise. We trained $\phi_{\tilde{\mathbf{w}}}$ on $\mathcal{D}_{\text{train}}^{\text{syn}}$ and tested it on $\mathcal{D}_{\text{test}}^{\text{syn}}$.

**Spurious bias.** We observe a high average accuracy of 97.4% but a WGA of 58.6% (Fig. 3(a), top) on the training data. The results show that the model heavily relies on the correlations that exist in the majority of samples and exhibits strong spurious bias. As expected, the performance on the test data is significantly lower (Fig. 3(a), bottom). The decision boundary (Fig. 3(a), green lines) learned from the training data does not generalize to the test data.

**Mitigation strategy.** Without group labels, it is challenging to identify and mitigate spurious bias captured by the model. We tackle this challenge by first finding that the distributions of values of an input dimension, together with the prediction outcomes for a certain class, provide discriminative information regarding the spuriousness of the dimension. (1) When the values for misclassified samples at the dimension are high, while values for the correctly predicted samples are low, this

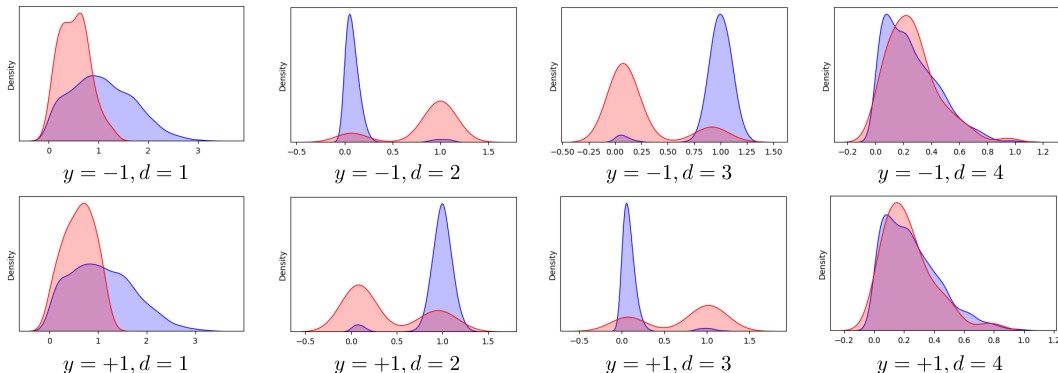

Figure 4: Distributions of values at all the four dimensions for the two classes -1 and +1 in the motivating example in Section A.1. "d=1" denotes the first dimension.

indicates that the absence of the dimension input does not significantly affect the correctness of predictions, while the presence of the dimension input does not generalize to certain groups of data. Therefore, the dimension tends to represent a spurious feature. For example, the center plot of Fig. 3(b) depicts the value distributions of the second dimension of input embeddings when $y_i = -1$. We obtain a similar plot for the third dimension of input embeddings when $y_i = +1$. (2) In contrast, if the absence of the dimension input results in misclassification, then the dimension tends to represent a core feature. The top plot of Fig. 3(b) represents the first dimension of input embeddings when $y_i = -1$. (3) For the noise dimension, i.e., the fourth dimension, due to randomness, there is little difference between the two distributions (Fig. 3(b) bottom). See Fig. 4 for all the plots. Next, we retrain the model while blocking the second, third, and fourth dimensions. As a result, the retrained model has learned to balance its performance on both the training and test data with a significant increase in WGA on the test data (Fig. 3(c)).

## A.2 THEORETICAL ANALYSIS

### A.2.1 PRELIMINARY

Based on the data model in Equation (9) and Equation (10), we restate the following

$$\mathbf{x} = (\mathbf{x}_{\text{core}}, \mathbf{x}_{\text{spu}})^T \in \mathbb{R}^{D \times 1}, \ y = \boldsymbol{\beta}^T \mathbf{x}_{\text{core}} + \varepsilon_{\text{core}}, \tag{13}$$

and

$$\mathbf{x}_{\text{spu}} = (2a - 1)\boldsymbol{\gamma} y + \boldsymbol{\varepsilon}_{\text{spu}}, a \sim \text{Bern}(p), \tag{14}$$

where $(2a - 1) \in \{-1, +1\}$, $a \sim \text{Bern}(p)$ is a Bernoulli random variable, $p$ is close to 1, $\varepsilon_{\text{core}}$ is a zero-mean Gaussian random variable with the variance $\eta_{\text{core}}^2$, and each element in $\boldsymbol{\varepsilon}_{\text{spu}}$ follows a zero-mean Gaussian distribution with the variance $\eta_{\text{spu}}^2$. We set $\eta_{\text{core}}^2 \gg \eta_{\text{spu}}^2$ to facilitate the learning of spurious features. The model $f(\mathbf{x}) = \mathbf{b}^T \mathbf{W} \mathbf{x}$ in Section 3.3 can be further expressed as follows,

$$\hat{y} = \sum_{i=1}^{M} b_i (\mathbf{x}_{\text{core}}^T \mathbf{w}_{\text{core},i} + \mathbf{x}_{\text{spu}}^T \mathbf{w}_{\text{spu},i}) = \mathbf{x}_{\text{core}}^T \mathbf{u}_{\text{core}} + \mathbf{x}_{\text{spu}}^T \mathbf{u}_{\text{spu}}, \tag{15}$$

where $\mathbf{w}_i^T \in \mathbb{R}^{1 \times D}$ is the $i$'th row of $\mathbf{W}$, $\mathbf{w}_i^T = [\mathbf{w}_{\text{core},i}^T, \mathbf{w}_{\text{spu},i}^T]$ with $\mathbf{w}_{\text{core},i} \in \mathbb{R}^{D_1 \times 1}$ and $\mathbf{w}_{\text{spu},i} \in \mathbb{R}^{D_2 \times 1}$, $\mathbf{u}_{\text{core}} = \sum_{i=1}^{M} b_i \mathbf{w}_{\text{core},i}$, and $\mathbf{u}_{\text{spu}} = \sum_{i=1}^{M} b_i \mathbf{w}_{\text{spu},i}$. The loss function which we use to optimize $\mathbf{W}$ and $\mathbf{b}$ is

$$\ell_{\text{tr}}(\mathbf{W}, \mathbf{b}) = \frac{1}{2} \mathbb{E}_{(\mathbf{x},y) \in \mathcal{D}_{\text{train}}} \|f(\mathbf{x}) - y\|_2^2. \tag{16}$$

With the above definitions, the following lemma gives the optimal coefficients $\mathbf{u}_{\text{core}}^*$ and $\mathbf{u}_{\text{spu}}^*$ based on the training data.

### A.2.2 PROOF FOR LEMMA 1

**Lemma 1.** *Given a training dataset $\mathcal{D}_{train}$ with $p$ defined in Equation (14) satisfying $1 \geq p \gg 0.5$, the optimized weights in the form of $\mathbf{u}^*_{core}$ and $\mathbf{u}^*_{spu}$ are*

$$\mathbf{u}^*_{core} = \frac{(2-2p)\eta^2_{core} + \eta^2_{spu}}{\eta^2_{core} + \eta^2_{spu}}\boldsymbol{\beta}, \tag{17}$$

*and*

$$\mathbf{u}^*_{spu} = \frac{(2p-1)\eta^2_{core}}{\eta^2_{core} + \eta^2_{spu}}\boldsymbol{\gamma}, \tag{18}$$

*respectively. When $p = 0.5$, the training data is unbiased and we obtain an unbiased classifier with weights $\mathbf{u}^*_{core} = \boldsymbol{\beta}$ and $\mathbf{u}^*_{spu} = 0$.*

*Proof.* Note that $f(\mathbf{x}) = \mathbf{b}^T\mathbf{W}\mathbf{x} = \mathbf{x}^T\mathbf{v} = \mathbf{x}^T_{\text{core}}\mathbf{u}_{\text{core}} + \mathbf{x}^T_{\text{spu}}\mathbf{u}_{\text{spu}}$, then we have

$$\ell_{\text{tr}}(W, b) = \frac{1}{2}\mathbb{E}\|\mathbf{x}^T_{\text{core}}\mathbf{u}_{\text{core}} + \mathbf{x}^T_{\text{spu}}\mathbf{u}_{\text{spu}} - y\|^2_2 \tag{19}$$

$$= \frac{1}{2}\mathbb{E}\|\mathbf{x}^T_{\text{core}}\mathbf{u}_{\text{core}} + \left[(2a-1)\boldsymbol{\gamma}y + \boldsymbol{\varepsilon}_{\text{spu}}\right]^T\mathbf{u}_{\text{spu}} - y\|^2_2 \tag{20}$$

$$= \frac{1}{2}\mathbb{E}\|\mathbf{x}^T_{\text{core}}\mathbf{u}_{\text{core}} - \left[1 - (2a-1)\boldsymbol{\gamma}^T\mathbf{u}_{\text{spu}}\right]y\|^2_2 + \frac{1}{2}\eta^2_{\text{spu}}\|\mathbf{u}_{\text{spu}}\|^2_2 \tag{21}$$

$$= \frac{1}{2}(pE_1 + (1-p)E_2) + \frac{1}{2}\eta^2_{\text{spu}}\|\mathbf{u}_{\text{spu}}\|^2_2, \tag{22}$$

where $E_1 = \|\mathbf{x}^T_{\text{core}}\mathbf{u}_{\text{core}} - (1 - \boldsymbol{\gamma}^T\mathbf{u}_{\text{spu}})y\|^2_2$ when $a = 1$ and $E_2 = \|\mathbf{x}^T_{\text{core}}\mathbf{u}_{\text{core}} - (1 + \boldsymbol{\gamma}^T\mathbf{u}_{\text{spu}})y\|^2_2$ when $a = 0$. We first calculate the lower bound for $E_1$ as follows

$$E_1 = \mathbb{E}\|\mathbf{x}^T_{\text{core}}\mathbf{u}_{\text{core}} - (1 - \boldsymbol{\gamma}^T\mathbf{u}_{\text{spu}})(\boldsymbol{\beta}^T\mathbf{x}_{\text{core}} + \varepsilon_{\text{core}})\|^2_2 \tag{23}$$

$$= \mathbb{E}\|\mathbf{x}^T_{\text{core}}\mathbf{u}_{\text{core}} - (1 - \boldsymbol{\gamma}^T\mathbf{u}_{\text{spu}})\boldsymbol{\beta}^T\mathbf{x}_{\text{core}} + (1 - \boldsymbol{\gamma}^T\mathbf{u}_{\text{spu}})\varepsilon_{\text{core}})\|^2_2 \tag{24}$$

$$= \mathbb{E}\|\mathbf{x}^T_{\text{core}}\mathbf{u}_{\text{core}} - (1 - \boldsymbol{\gamma}^T\mathbf{u}_{\text{spu}})\boldsymbol{\beta}^T\mathbf{x}_{\text{core}}\|^2_2 + \eta^2_{\text{core}}(1 - \boldsymbol{\gamma}^T\mathbf{u}_{\text{spu}})^2 \tag{25}$$

$$\geq \eta^2_{\text{core}}(1 - \boldsymbol{\gamma}^T\mathbf{u}_{\text{spu}})^2. \tag{26}$$

Similarly, we have

$$E_2 = \mathbb{E}\|\mathbf{x}^T_{\text{core}}\mathbf{u}_{\text{core}} - (1 + \boldsymbol{\gamma}^T\mathbf{u}_{\text{spu}})(\boldsymbol{\beta}^T\mathbf{x}_{\text{core}} + \varepsilon_{\text{core}})\|^2_2 \tag{27}$$

$$= \mathbb{E}\|\mathbf{x}^T_{\text{core}}\mathbf{u}_{\text{core}} - (1 + \boldsymbol{\gamma}^T\mathbf{u}_{\text{spu}})\boldsymbol{\beta}^T\mathbf{x}_{\text{core}}\|^2_2 + \eta^2_{\text{core}}(1 + \boldsymbol{\gamma}^T\mathbf{u}_{\text{spu}})^2 \tag{28}$$

$$\geq \eta^2_{\text{core}}(1 + \boldsymbol{\gamma}^T\mathbf{u}_{\text{spu}})^2. \tag{29}$$

Then, plug in (26) and (29) into (22), we obtain the following

$$\ell_{\text{tr}}(W, b) \geq \frac{1}{2}\left(p\eta^2_{\text{core}}(1 - \boldsymbol{\gamma}^T\mathbf{u}_{\text{spu}})^2 + (1-p)\eta^2_{\text{core}}(1 + \boldsymbol{\gamma}^T\mathbf{u}_{\text{spu}})^2 + \eta^2_{\text{spu}}\|\mathbf{u}_{\text{spu}}\|^2_2\right) \tag{30}$$

$$= \frac{1}{2}\left(p\eta^2_{\text{core}}(1 - \boldsymbol{\gamma}^T\mathbf{u}_{\text{spu}})^2 + (1-p)\eta^2_{\text{core}}(1 + \boldsymbol{\gamma}^T\mathbf{u}_{\text{spu}})^2 + \eta^2_{\text{spu}}\|\boldsymbol{\gamma}\|^2_2\|\mathbf{u}_{\text{spu}}\|^2_2\right) \tag{31}$$

$$\geq \frac{1}{2}\left(p\eta^2_{\text{core}}(1 - \boldsymbol{\gamma}^T\mathbf{u}_{\text{spu}})^2 + (1-p)\eta^2_{\text{core}}(1 + \boldsymbol{\gamma}^T\mathbf{u}_{\text{spu}})^2 + \eta^2_{\text{spu}}\|\boldsymbol{\gamma}^T\mathbf{u}_{\text{spu}}\|^2_2\right), \tag{32}$$

where Equation (31) uses the fact that $\boldsymbol{\gamma}$ has a unit norm, and the inequality (32) exploits the Cauchy–Schwarz inequality. Let $z = \boldsymbol{\gamma}^T\mathbf{u}_{\text{spu}}$, we have $\ell(z) = p\eta^2_{\text{core}}(1 - z)^2 + (1 - p)\eta^2_{\text{core}}(1 + z)^2 + \eta^2_{\text{spu}}z^2$. Let $\frac{\partial\ell(z)}{\partial z} = 0$, we obtain

$$z^* = \boldsymbol{\gamma}^T\mathbf{u}^*_{\text{spu}} = \frac{(2p-1)\eta^2_{\text{core}}}{\eta^2_{\text{core}} + \eta^2_{\text{spu}}}.$$

Given $\mathbf{u}^*_{\text{spu}}$, we can obtain the optimal $\mathbf{u}'_{\text{core}}$ for minimizing $E_1$ in Equation (25) as $\mathbf{u}'_{\text{core}} = (1 - z^*)\boldsymbol{\beta}$; similarly, we can obtain the optimal $\mathbf{u}''_{\text{core}}$ for minimizing $E_2$ in Equation (28) as $\mathbf{u}''_{\text{core}} = (1 + z^*)\boldsymbol{\beta}$. Via proof by contradiction, only $\mathbf{u}'_{\text{core}}$ or $\mathbf{u}''_{\text{core}}$ is the solution for $\mathbf{u}^*$core. Since $p \gg 0.5$, $E_1$ contributes to the majority error. Thus, $\mathbf{u}^*_{\text{core}} = (1 - z^*)\boldsymbol{\beta}$, i.e.,

$$\mathbf{u}^*_{\text{core}} = (1 - z^*)\boldsymbol{\beta} = \frac{(2-2p)\eta^2_{\text{core}} + \eta^2_{\text{spu}}}{\eta^2_{\text{core}} + \eta^2_{\text{spu}}}\boldsymbol{\beta}.$$

$\square$

### A.2.3 PROOF FOR COROLLARY 1

Lemma 1 gives the optimal model weights under a given training dataset $\mathcal{D}_{\text{train}}$ with the parameter $p$ controlling the strength of spurious correlations. Lemma 1 generalizes the result in Ye et al. (2023) where $p = 1$. Importantly, we obtain the following corollary for unbiased models:

**Corollary 1.** *The unbiased model* $f(\mathbf{x}) = \mathbf{u}^T\mathbf{x} = \mathbf{x}_{core}^T\mathbf{u}_{core} + \mathbf{x}_{spu}^T\mathbf{u}_{spu}$ *is achieved when* $\mathbf{u}_{core} = \mathbf{u}_{core}^*$ *and* $\boldsymbol{\gamma}^T\mathbf{u}_{spu} = 0$.

*Proof.* Plug $\boldsymbol{\gamma}^T\mathbf{u}_{\text{core}} = 0$ into Equation (25) and Equation (28), then we observe that $\mathbf{u}_{\text{core}}$ minimizes errors from both the majority ($a = 1$) and minority ($a = 0$) groups of data. $\square$

If we could obtain a set of unbiased training data with $p = 0.5$, then we obtain an unbiased model with $\mathbf{u}_{\text{spu}}^* = 0$ and $\mathbf{u}_{\text{core}}^* = \boldsymbol{\beta}$. However, in practice, it is challenging to obtain a set of unbiased training data, i.e., it is challenging to control the value of $p$.

### A.2.4 PROOF FOR PROPOSITION 1

**Proposition 1** (**Principal for selective activation**). *Given the model* $f(\mathbf{x}) = \mathbf{b}^T\mathbf{W}\mathbf{x}$ *trained with data generated under the data model specified in Equation (13) and Equation (14), it captures spurious correlations when* $\boldsymbol{\gamma}^T\mathbf{w}_{spu,i} < 0, i \in \{1, \ldots, M\}$. *The principal of selective activation is to mask out neurons containing negative* $\boldsymbol{\gamma}^T\mathbf{w}_{spu,i}$.

*Proof.* Consider the $i$'th neuron $e_i$ ($i = 1, \ldots, M$) before the last layer. We first expand it based on our data model specified by Equation (13) and Equation (14) as follows:

$$e_i = \mathbf{x}_{\text{core}}^T\mathbf{w}_{\text{core},i} + \mathbf{x}_{\text{spu}}^T\mathbf{w}_{\text{spu},i} \tag{33}$$

$$= \mathbf{x}_{\text{core}}^T\mathbf{w}_{\text{core},i} + [(2a-1)\boldsymbol{\gamma}y + \boldsymbol{\varepsilon}_{\text{spu}}]^T\mathbf{w}_{\text{spu},i} \tag{34}$$

$$= \mathbf{x}_{\text{core}}^T\mathbf{w}_{\text{core},i} + (2a-1)[\boldsymbol{\beta}^T\mathbf{x}_{\text{core}} + \varepsilon_{\text{core}}]\boldsymbol{\gamma}^T\mathbf{w}_{\text{spu},i} + \boldsymbol{\varepsilon}_{\text{spu}}^T\mathbf{w}_{\text{spu},i} \tag{35}$$

$$= \mathbf{x}_{\text{core}}^T\mathbf{w}_{\text{core},i} + (2a-1)\boldsymbol{\beta}^T\mathbf{x}_{\text{core}}\boldsymbol{\gamma}^T\mathbf{w}_{\text{spu},i} + \varepsilon_{\text{rem}}, \tag{36}$$

where $\varepsilon_{\text{rem}} = \varepsilon_{\text{core}}\boldsymbol{\gamma}^T\mathbf{w}_{\text{spu},i} + \boldsymbol{\varepsilon}_{\text{spu}}^T\mathbf{w}_{\text{spu},i}$. In Equation (36), if $\boldsymbol{\gamma}^T\mathbf{w}_{\text{spu},i} \geq 0$, the model handles the spurious component correctly. Specifically, when $a = 1$, the spurious component positively correlates with the core component and contributes to the output, whereas when $a = 0$, its correlation with the core component breaks with a negative one and has a negative contribution to the output. In contrast, if $\boldsymbol{\gamma}^T\mathbf{w}_{\text{spu},i} < 0$ and $a = 1$, then the model still utilizes the spurious component even the correlation breaks, demonstrating a strong reliance on the spurious component instead of the core component. Therefore, the principal of selective activation is to find neurons containing negative $\boldsymbol{\gamma}^T\mathbf{w}_{\text{spu},i}$ so that masking them out improves the model's generalization. $\square$

### A.2.5 PROOF FOR THEOREM 1

The following theorem validates our neuron selection method.

**Theorem 1** (**Metric for neuron selection**). *Given the model* $f(\mathbf{x}) = \mathbf{b}^T\mathbf{W}\mathbf{x}$, *we cast it to a classification model by training it to regress* $y \in \{-\mu, \mu\}$ ($\mu > 0$) *on* $\mathbf{x}$ *based on the data model specified in Equation (13) and Equation (14), where* $\mu = \mathbb{E}[\boldsymbol{\beta}^T\mathbf{x}_{core}]$. *The metric* $\delta_i^y$ *defined in the following can identify neurons with spurious correlations when* $\delta_i^y > 0$:

$$\delta_i^y = Med(\bar{\mathcal{V}}_i^y) - Med(\hat{\mathcal{V}}_i^y),$$

*where* $\bar{\mathcal{V}}_i^y$ *and* $\hat{\mathcal{V}}_i^y$ *are the sets of activation values for misclassified and correctly predicted samples with the label* $y$ *from the* $i$'th *neuron, respectively;* $Med(\cdot)$ *denotes the Median operator; and an activation value is defined as* $\mathbf{x}_{core}^T\mathbf{w}_{core,i} + \mathbf{x}_{spu}^T\mathbf{w}_{spu,i}$.

*Proof.* We start by obtaining the set of correctly predicted samples $\hat{\mathcal{D}}_y$ and the set of incorrectly predicted samples $\bar{\mathcal{D}}_y$ as $\hat{\mathcal{D}}_y = \{\mathbf{x}|f(\mathbf{x}) \geq 0, (\mathbf{x}, y) \in \mathcal{D}_{\text{Ide}}\}$ and $\bar{\mathcal{D}}_y = \{\mathbf{x}|f(\mathbf{x}) < 0, (\mathbf{x}, y) \in \mathcal{D}_{\text{Ide}}\}$, where $\mathcal{D}_{\text{Ide}}$ is the set of identification data. Then, we have $\hat{\mathcal{V}}_i^y = \{e_i|\mathbf{x} \in \hat{\mathcal{D}}_y\}$, and $\bar{\mathcal{V}}_i^y = \{e_i|\mathbf{x} \in$

$\bar{\mathcal{D}}_y\}$, where $e_i$ is the $i$'th neuron activation defined in Equation (36). Expanding $e_i$ following Equation (36), we obtain

$$e_i = \mathbf{x}_{\text{core}}^T \mathbf{w}_{\text{core},i} + (2a-1)\beta^T \mathbf{x}_{\text{core}} \gamma^T \mathbf{w}_{\text{spu},i} + \varepsilon_{\text{rem}}.$$

Note that $\mathbf{x}_{\text{core}}^T \mathbf{w}_{\text{core},i}$ and $\varepsilon_{\text{rem}}$ exist for all the samples, regardless of the ultimate prediction results, and all $e_i$ follows a Gaussian distribution given $a$. Then, among all the correctly predicted samples with the label $y$, according the Lemma 2, we have $\text{Med}(\hat{\mathcal{V}}_i^y) \approx \mathbb{E}[\mathbf{x}_{\text{core}}^T \mathbf{w}_{\text{core},i}] + \mu\gamma^T \mathbf{w}_{\text{spu},i}$. Similarly, among all the incorrectly predicted samples with the label $y$, we have $\text{Med}(\bar{\mathcal{V}}_i^y) \approx \mathbb{E}[\mathbf{x}_{\text{core}}^T \mathbf{w}_{\text{core},i}] - \mu\gamma^T \mathbf{w}_{\text{spu},i}$. Then, the difference between the two is

$$\delta_i^y \approx -2\mu\gamma^T \mathbf{w}_{\text{spu},i}.$$

When $\delta_i^y > 0$, we have $\gamma^T \mathbf{w}_{\text{spu},i} < 0$. According Proposition 1, using $\delta_i^y > 0$ indeed selects neurons that have strong reliance on spurious components. $\qquad\square$

### A.2.6 PROOF FOR THEOREM 2

**Theorem 2 (LaSAR mitigates spurious bias).** *Consider the model $f^*(\mathbf{x}) = \mathbf{x}^T \mathbf{u}^*$ trained on the biased training data with $p \gg 0.5$, with $\mathbf{u}_{core}^*$ and $\mathbf{u}_{spu}^*$ defined in Equation (17) and Equation (18), respectively. Under the mild assumption that $\beta^T \mathbf{w}_{core,i} \approx \gamma^T \mathbf{w}_{spu,i}, \forall i = 1, \ldots, M$, then applying LaSAR to $f^*(\mathbf{x})$ produces a model that is closer to the unbiased one.*

*Proof.* Consider $f^*(\mathbf{x})$ as the base model. We aim to prove that the retrained model obtained with LaSAR produces model parameters that is closer to the unbiased model defined in Corollary 1 than the base model.

First, the assumption that $\beta^T \mathbf{w}_{\text{core},i} \approx \gamma^T \mathbf{w}_{\text{spu},i}, \forall i = 1, \ldots, M$ generally holds for a biased model as the model has learned to associate spurious features with the core features.

Then, we denote the retrained parameters obtained with LaSAR as $\mathbf{u}_{\text{core}}^\dagger$ and $\mathbf{u}_{\text{spu}}^\dagger$. We start with calculating $\mathbf{u}_{\text{spu}}^\dagger$. Focusing on Equation (32) and following the derivation in Lemma 1, we obtain $\mathbf{u}_{\text{spu}}^\dagger = \sum_{i \in \mathcal{I}_+} b_i \mathbf{w}_{\text{spu},i} = \mathbf{u}_{\text{spu}}^*$, where $\mathcal{I}_+$ denotes the set of neuron indexes satisfying $\gamma^T \mathbf{w}_{\text{spu},i} > 0$. Note that LaSAR is a last-layer retraining method; thus we only optimize $b_i$ here and $\mathbf{w}_{\text{spu},i}$ is the same as in $f^*(\mathbf{x})$. Left multiplying $\mathbf{u}_{\text{spu}}^\dagger$ with $\gamma^T$, we have

$$\gamma^T \mathbf{u}_{\text{spu}}^\dagger = \sum_{i \in \mathcal{I}_+} b_i^\dagger \gamma^T \mathbf{w}_{\text{spu},i} \tag{37}$$

$$= z^* = \frac{(2p-1)\eta_{\text{core}}^2}{\eta_{\text{core}}^2 + \eta_{\text{spu}}^2} > 0.$$

Note that $\gamma^T \mathbf{w}_{\text{spu},i} > 0$, $\forall i \in \mathcal{I}_+$ because of LaSAR. Hence, we have $b_i^\dagger > 0$, $\forall i \in \mathcal{I}_+$. Moreover, we observe that $\mathbf{u}_{\text{spu}}^\dagger$ is the same as $\mathbf{u}_{\text{spu}}^*$ as long as $\mathcal{I}_+$ is non-empty. This shows that LaSAR is not able to optimize parameters related to the spurious components in the input data.

According to the Corollary 1, the unbiased model is achieved when $p = 0.5$ and $\mathbf{u}_{\text{core}} = \beta$. The Euclidean distance between $\beta$ and the biased solution $\mathbf{u}_{\text{core}} = (1 - z^*)\beta$ is $\|\mathbf{u}_{\text{core}}^* - \beta\| = z^*$. Based on Equation (37), we estimate the distance between our LaSAR solution $\mathbf{u}_{\text{core}}^\dagger$ and $\beta$ as follows

$$\|\mathbf{u}_{\text{core}}^\dagger - \beta\|_2 = \|\beta^T(\mathbf{u}_{\text{core}}^\dagger - \beta)\|_2 \tag{38}$$

$$= \|\beta^T \mathbf{u}_{\text{core}}^\dagger - 1\|_2 \tag{39}$$

$$= \|\sum_{i \in \mathcal{I}_+} b_i^\dagger \beta^T \mathbf{w}_{\text{core},i} - 1\|_2 \tag{40}$$

$$\approx \|\sum_{i \in \mathcal{I}_+} b_i^\dagger \gamma^T \mathbf{w}_{\text{spu},i} - 1\|_2 \tag{41}$$

$$= \|z^* - 1\|, \tag{42}$$

where Equation (39) uses the fact that $\boldsymbol{\beta}^T\boldsymbol{\beta} = 1$, and Equation (40) uses the condition $\boldsymbol{\beta}^T\mathbf{w}_{\text{core},i} \approx \boldsymbol{\gamma}^T\mathbf{w}_{\text{spu},i}, \forall i = 1, \ldots, M$. Note that $z^*$ is achieved on the training data with $p \gg 0.5$ and $\eta_{\text{core}}^2 \gg \eta_{\text{spu}}^2$, hence we have $z^* \approx 1$ and $\|\mathbf{u}_{\text{core}}^\dagger - \boldsymbol{\beta}\|_2 \approx 0$. In other words, LaSAR can bring model parameters closer to the optimal and unbiased solution than the parameters of the biased model.

$\square$

### A.2.7  PROOF FOR LEMMA 2

**Lemma 2** (**Majority of samples among different predictions**). *Given the model $f(\mathbf{x}) = \mathbf{b}^T\mathbf{W}\mathbf{x}$ trained on $y \in \{-\mu, \mu\}$ ($\mu > 0$) with $\mu = \mathbb{E}[\boldsymbol{\beta}^T\mathbf{x}_{core}]$, and the conditions that $p > 3/4$ and $\eta_{core}^2 \gg \eta_{spu}^2$, we have the following claims:*

- *Among the set of all correctly predicted samples with the label $y$, more than half of them are generated with $a = 1$;*

- *Among the set of all incorrectly predicted samples with the label $y$, more than half of them are generated with $a = 0$.*

*Proof.* With the two regression targets, $-\mu$ and $\mu$, the optimal decision boundary is 0. Without loss of generality, we consider $y = \mu$. Then, the set of correctly predicted samples $\hat{\mathcal{D}}_y$ is

$$\hat{\mathcal{D}}_y = \{\mathbf{x}|f(\mathbf{x}) \geq 0, (\mathbf{x}, y) \in \mathcal{D}_{\text{Ide}}\},$$

and the set of incorrectly predicted samples $\hat{\mathcal{D}}_y$ is

$$\bar{\mathcal{D}}_y = \{\mathbf{x}|f(\mathbf{x}) < 0, (\mathbf{x}, y) \in \mathcal{D}_{\text{Ide}}\}.$$

The probability of a sample with the label $y$ that is correctly predicted is

$$P(\mathbf{x} \in \hat{\mathcal{D}}_y|y) = P(a = 1)P(f(\mathbf{x}) \geq 0|a = 1, y) + P(a = 0)P(f(\mathbf{x}) \geq 0|a = 0, y)$$
$$= pP(f(\mathbf{x}) \geq 0|a = 1, y) + (1 - p)P(f(\mathbf{x}) \geq 0|a = 0, y).$$

Similarly, the probability of a sample with the label $y$ that is incorrectly predicted is

$$P(\mathbf{x} \in \bar{\mathcal{D}}_y|y) = pP(f(\mathbf{x}) < 0|a = 1, y) + (1 - p)P(f(\mathbf{x}) < 0|a = 0, y).$$

To calculate $P(f(\mathbf{x}) \geq 0|a = 1, y)$, we expand $f(\mathbf{x})$ as follows:

$$f(\mathbf{x}) = \mathbf{x}_{\text{core}}^T\mathbf{u}_{\text{core}}^* + \mathbf{x}_{\text{spu}}^T\mathbf{u}_{\text{spu}}^*$$
$$= \mathbf{x}_{\text{core}}^T\boldsymbol{\beta}(1 - z^*) + (\boldsymbol{\gamma}(\boldsymbol{\beta}^T\mathbf{x}_{\text{core}} + \varepsilon_{\text{core}}) + \boldsymbol{\varepsilon}_{\text{spu}})^T\mathbf{u}_{\text{spu}}^*$$
$$= \mathbf{x}_{\text{core}}^T\boldsymbol{\beta}(1 - z^*) + \mathbf{x}_{\text{core}}^T\boldsymbol{\beta}\boldsymbol{\gamma}^T\mathbf{u}_{\text{spu}}^* + \boldsymbol{\gamma}^T\mathbf{u}_{\text{spu}}^*\varepsilon_{\text{core}} + \boldsymbol{\varepsilon}_{\text{spu}}^T\mathbf{u}_{\text{spu}}^*$$
$$= \mathbf{x}_{\text{core}}^T\boldsymbol{\beta} + z^*\varepsilon_{\text{core}} + \boldsymbol{\varepsilon}_{\text{spu}}^T\mathbf{u}_{\text{spu}}^*$$

The output of $f(\mathbf{x})$ follows a Gaussian distribution, with the mean $\mu_1 = \mathbb{E}[f(\mathbf{x})] = \mu$, and the variance $\sigma_1^2 = Var(\mathbf{x}_{\text{core}}^T\boldsymbol{\beta}) + \eta_{\text{core}}^2(z^*)^2 + \eta_{\text{spu}}^2(z^*)^2$. Therefore, we have

$$P(f(\mathbf{x}) \geq 0|a = 1, y) = P(\mathbf{x} \in \hat{\mathcal{D}}_y|a = 1, y) = 1 - \Phi(\frac{0 - \mu}{\sigma_1}) = \Phi(\frac{\mu}{\sigma_1}), \quad (43)$$

$$P(f(\mathbf{x}) < 0|a = 1, y) = P(\mathbf{x} \in \bar{\mathcal{D}}_y|a = 1, y) = 1 - \Phi(\frac{\mu}{\sigma_1}) = \Phi(\frac{-\mu}{\sigma_1}). \quad (44)$$

Similarly, to calculate $P(f(\mathbf{x}) \geq 0|a = 0, y)$, we expand $f(\mathbf{x})$ as follows:

$$f(\mathbf{x}) = \mathbf{x}_{\text{core}}^T\boldsymbol{\beta}(1 - z^*) - \mathbf{x}_{\text{core}}^T\boldsymbol{\beta}\boldsymbol{\gamma}^T\mathbf{u}_{\text{spu}}^* - \boldsymbol{\gamma}^T\mathbf{u}_{\text{spu}}^*\varepsilon_{\text{core}} + \boldsymbol{\varepsilon}_{\text{spu}}^T\mathbf{u}_{\text{spu}}^*$$
$$= \mathbf{x}_{\text{core}}^T\boldsymbol{\beta}(1 - 2z^*) - z^*\varepsilon_{\text{core}} + \boldsymbol{\varepsilon}_{\text{spu}}^T\mathbf{u}_{\text{spu}}^*.$$

The output of $f(\mathbf{x})$ follows a Gaussian distribution, with the mean $\mu_0 = \mathbb{E}[f(\mathbf{x})] = \mu(1 - 2z^*)$, and the variance $\sigma_0^2 = (1 - 2z^*)^2 Var(\mathbf{x}_{\text{core}}^T\boldsymbol{\beta}) + \eta_{\text{core}}^2(z^*)^2 + \eta_{\text{spu}}^2(z^*)^2$. Therefore, we have

$$P(f(\mathbf{x}) \geq 0|a = 0, y) = P(x \in \hat{\mathcal{D}}_y|a = 0, y) = 1 - \Phi(\frac{0 - \mu_0}{\sigma_0}) = \Phi(\frac{(1 - 2z^*)\mu}{\sigma_0}), \quad (45)$$

$$P(f(\mathbf{x}) < 0|a = 0, y) = P(x \in \bar{\mathcal{D}}_y|a = 0, y) = 1 - \Phi(\frac{\mu_0}{\sigma_0}) = \Phi(\frac{-(1 - 2z^*)\mu}{\sigma_0}). \quad (46)$$

Therefore, we have the probabilities for correctly and incorrectly predicted samples with the label $y$, i.e.,

$$P(\mathbf{x} \in \hat{\mathcal{D}}_y | y) = p\Phi(\frac{\mu}{\sigma_1}) + (1-p)\Phi(\frac{(1-2z^*)\mu}{\sigma_0}), \tag{47}$$

and

$$P(\mathbf{x} \in \bar{\mathcal{D}}_y | y) = p\Phi(\frac{-\mu}{\sigma_1}) + (1-p)\Phi(\frac{-(1-2z^*)\mu}{\sigma_0}) \tag{48}$$

Next, we seek to determine whether the majority of samples in the correctly (incorrectly) predicted set $\hat{\mathcal{D}}_y$ ($\bar{\mathcal{D}}_y$) is generated with $a = 0$ or $a = 1$. To achieve this, in the set of correctly predicted samples, we use the Bayesian theorem based on Equation (47), i.e.,

$$P(a = 1 | \mathbf{x} \in \hat{\mathcal{D}}_y, y) = \frac{P(\mathbf{x} \in \hat{\mathcal{D}}_y | a = 1, y)P(a = 1)}{P(\mathbf{x} \in \hat{\mathcal{D}}_y | y)}$$
$$= \frac{p\Phi(\mu/\sigma_1)}{p\Phi(\mu/\sigma_1) + (1-p)\Phi((1-2z^*)\mu/\sigma_0)}, \tag{49}$$

and

$$P(a = 0 | \mathbf{x} \in \hat{\mathcal{D}}_y, y) = 1 - P(a = 1 | \mathbf{x} \in \hat{\mathcal{D}}_y, y)$$
$$= \frac{(1-p)\Phi((1-2z^*)\mu/\sigma_0)}{p\Phi(\mu/\sigma_1) + (1-p)\Phi((1-2z^*)\mu/\sigma_0)}. \tag{50}$$

Similarly, in the set of incorrectly predicted samples, we have

$$P(a = 1 | \mathbf{x} \in \bar{\mathcal{D}}_y, y) = \frac{P(\mathbf{x} \in \bar{\mathcal{D}}_y | a = 1, y)P(a = 1)}{P(\mathbf{x} \in \bar{\mathcal{D}}_y | y)}$$
$$= \frac{p\Phi(-\mu/\sigma_1)}{p\Phi(-\mu/\sigma_1) + (1-p)\Phi(-(1-2z^*)\mu/\sigma_0)}, \tag{51}$$

and

$$P(a = 0 | \mathbf{x} \in \bar{\mathcal{D}}_y, y) = 1 - P(a = 1 | \mathbf{x} \in \bar{\mathcal{D}}_y, y)$$
$$= \frac{(1-p)\Phi(-(1-2z^*)\mu/\sigma_0)}{p\Phi(-\mu/\sigma_1) + (1-p)\Phi(-(1-2z^*)\mu/\sigma_0)}. \tag{52}$$

Under the assumption that $p > 3/4$ and $\eta_{\text{core}}^2 \gg \eta_{\text{spu}}^2$, we have $1-2z^* = \left((3-4p)\eta_{\text{core}}^2 + \eta_{\text{spu}}^2\right)/(\eta_{\text{core}}^2 + \eta_{\text{spu}}^2) < 0$. Hence, $\Phi(-(1-2z^*)\mu/\sigma_0) < 1/2$ and $P(a = 1 | \mathbf{x} \in \hat{\mathcal{D}}_y, y) > 1/2$; in other words, **among the set of all correctly predicted samples with the label $y$, more than half of them are generated with $a = 1$.**

Moreover, under the assumption that $\Phi(-\mu/\sigma_1) \approx 0$, i.e., predictions of the model have a high signal-to-noise ratio, then $P(a = 0 | \mathbf{x} \in \bar{\mathcal{D}}_y, y) > 1/2$, i.e., **among the set of all incorrectly predicted samples with the label $y$, more than half of them are generated with $a = 0$.** This assumption is generally true, as $\sigma_1^2 = Var(\mathbf{x}_{\text{core}}^T \boldsymbol{\beta}) + \eta_{\text{core}}^2(z^*)^2 + \eta_{\text{spu}}^2(z^*)^2$ is typically very small when $z^*$ approaches zero given $p > 3/4$ and $\eta_{\text{core}}^2 \gg \eta_{\text{spu}}^2$. $\qquad\square$

### A.2.8 PROOF FOR LEMMA 3

**Lemma 3.** *Consider the model $f(\mathbf{x}) = \mathbf{x}^T \mathbf{u}$ with $\mathbf{u} = [\mathbf{u}_{core}, \mathbf{u}_{spu}]$, the optimal solution for $\mathbf{u}_{spu}$ that can be achieved by last-layer retraining on the retraining data with $p_{re}$ is $\mathbf{u}_{spu}^r$, which is defined as*

$$\mathbf{u}_{spu}^r = \frac{(2p_{re} - 1)\eta_{core}^2}{\eta_{core}^2 + \eta_{spu}^2}\boldsymbol{\gamma}. \tag{53}$$

*Proof.* First, we have $f(\mathbf{x}) = \mathbf{x}^T \mathbf{u} = \mathbf{b}^T \mathbf{W} \mathbf{x}$. For last-layer retraining, $\mathbf{b}$ is optimized. Following the derivation in Lemma 1, we similarly obtain the inequality in (32) with $p = p_{\text{re}}$, i.e.,

$$\ell(\mathbf{b}) \geq \frac{1}{2}\Big(p_{\text{re}}\eta_{\text{core}}^2(1 - \boldsymbol{\gamma}^T \mathbf{u}_{\text{spu}})^2 + (1-p_{\text{re}})\eta_{\text{core}}^2(1 + \boldsymbol{\gamma}^T \mathbf{u}_{\text{spu}})^2 + \eta_{\text{spu}}^2 \|\boldsymbol{\gamma}^T \mathbf{u}_{\text{spu}}\|_2^2\Big), \tag{54}$$

Note that the terms on the right side of the inequality are independent of any manipulation of the retraining data, such as reweighting. Then, taking the derivative to the sum of these terms with respect to $\mathbf{b}$, we obtain the following equation

$$\boldsymbol{\gamma}^T \mathbf{W}_{\text{spu}} \mathbf{b} = \frac{(2p_{\text{re}} - 1)\eta_{\text{core}}^2}{\eta_{\text{core}}^2 + \eta_{\text{spu}}^2}, \tag{55}$$

where $\mathbf{u}_{\text{spu}} = \mathbf{W}_{\text{spu}} \mathbf{b}$. Since $\boldsymbol{\gamma}^T \boldsymbol{\gamma} = 1$, then we have $\mathbf{u}_{\text{spu}} = \mathbf{u}_{\text{spu}}^r$. We finally verify that $\mathbf{u}_{\text{spu}}^r$ indeed minimizes the sum of the terms on the right hand side of (54). If $p_{\text{re}}$ equals to $p$ for the training data, then $\mathbf{u}_{\text{spu}}^r = \mathbf{u}_{\text{spu}}^*$ defined in Equation (18). $\square$

### A.3 CONNECTION TO LAST-LAYER RETRAINING METHODS

Although at the surface level, our method shares a similar setting to last-layer retraining methods, such as AFR (Qiu et al., 2023) and DFR (Kirichenko et al., 2023), our method is fundamentally different from these methods in how spurious bias is mitigated. Take AFR for an example. It, in essence, is a sample-level method and adjusts the weights of the last layer indirectly via retraining on samples with loss-related weights. Our method directly forces the weights identified as affected by spurious bias to zero, while adjusting the remaining weights with retraining.

The advantage of LaSAR can be explained more formally in our theoretical analysis framework. First, consider the training loss in Equation (22), we can express it as the sum of following terms for brevity,

$$\ell_{tr}(\mathbf{W}, \mathbf{b}) = \frac{1}{2} p \mathbb{E}[\psi_1(\mathbf{u}_{\text{core}}, \mathbf{u}_{\text{spu}}) =] + \frac{1}{2}(1-p)\mathbb{E}[\psi_2(\mathbf{u}_{\text{core}}, \mathbf{u}_{\text{spu}})] + \frac{1}{2}\psi_3(\mathbf{u}_{\text{spu}}), \tag{56}$$

where $p$ is the data generation parameter and is fixed, and $\psi_1$, $\psi_2$, and $\psi_3$ are defined as

$$\psi_1(\mathbf{u}_{\text{core}}, \mathbf{u}_{\text{spu}}) = \mathbb{E}\|\mathbf{x}_{\text{core}}^T \mathbf{u}_{\text{core}} - (1 - \boldsymbol{\gamma}^T \mathbf{u}_{\text{spu}})\boldsymbol{\beta}^T \mathbf{x}_{\text{core}}\|_2^2,$$

$$\psi_2(\mathbf{u}_{\text{core}}, \mathbf{u}_{\text{spu}}) = \mathbb{E}\|\mathbf{x}_{\text{core}}^T \mathbf{u}_{\text{core}} - (1 + \boldsymbol{\gamma}^T \mathbf{u}_{\text{spu}})\boldsymbol{\beta}^T \mathbf{x}_{\text{core}}\|_2^2,$$

and

$$\psi_3(\mathbf{u}_{\text{spu}}) = p\eta_{\text{core}}^2(1 - \boldsymbol{\gamma}^T \mathbf{u}_{\text{spu}})^2 + (1-p)\eta_{\text{core}}^2(1 + \boldsymbol{\gamma}^T \mathbf{u}_{\text{spu}})^2 + \eta_{\text{spu}}^2\|\boldsymbol{\gamma}^T \mathbf{u}_{\text{spu}}\|_2^2,$$

respectively. Based on Lemma 3, for last-layer retraining methods in general, the optimal solution for $\mathbf{u}_{\text{spu}}$ is $\mathbf{u}_{\text{spu}}^*$, given that the retraining data follows the same distribution as the training data.

AFR changes the distribution within the first two expectation terms $\psi_1(\mathbf{u}_{\text{core}}, \mathbf{u}_{\text{spu}})$ and $\psi_2(\mathbf{u}_{\text{core}}, \mathbf{u}_{\text{spu}})$ and jointly updates $\mathbf{u}_{\text{core}}$ and $\mathbf{u}_{\text{spu}}$, while there is no optimality guarantee for $\mathbf{u}_{\text{spu}}$ ($\psi_3(\mathbf{u}_{\text{spu}})$ is not considered in AFR). By contrast, according to Theorem 2, LaSAR first ensures that $\mathbf{u}_{\text{spu}}$ is optimal, then it moves $\mathbf{u}_{\text{core}}$ close the the unbiased solution.

### A.4 COMPLEXITY ANALYSIS

We analyze the computational complexity of our method, LaSAR, alongside representative reweighting-based methods, including AFR (Qiu et al., 2023), DFR (Kirichenko et al., 2023), and JTT (Liu et al., 2021). Let the number of identification samples be $N_{\text{Ide}}$, the number of retraining samples be $N_{\text{ret}}$, the total number of training samples be $N$, the number of latent dimensions be $D$, and the number of training epochs be $E$. Additionally, denote the time required for inference as $\tau_{\text{fw}}$, for last-layer retraining as $\tau_{\text{ll}}$, and for optimizing the entire model as $\tau_{\text{opt}}$. The computational complexities of these methods are summarized in Table 6.

Among the methods, JTT has the highest computational complexity since $\tau_{\text{opt}} \gg \tau_{\text{ll}}$, requiring full model optimization. DFR is much faster due to its reliance on last-layer retraining, though it requires group annotations. AFR extends DFR by additionally precomputing sample losses, increasing its computational cost slightly. LaSAR, while requiring more time than AFR to identify spurious dimensions across all $D$ embedding dimensions, remains computationally efficient. This is because $\tau_{\text{fw}}$, the time required for forward inference, is typically very small. As a result, LaSAR offers an effective balance between computational efficiency and robust spurious bias mitigation.

## A.5 ADVANTAGES OVER VARIABLE SELECTION METHODS

Although the identification of spurious dimensions in Equation (6) may resemble traditional variable selection methods (Heinze et al., 2018), our approach extends beyond simply selecting a subset of variables that optimally explain the target variable. Instead, it specifically addresses spurious bias—an issue often neglected in traditional variable selection.

Traditional variable selection methods, such as L1 regularization, do not distinguish whether variables represent spurious or core features. Since spurious features are often predictive of target labels in the training data and are easier for models to learn (Tiwari & Shenoy, 2023; Ye et al., 2023), these methods may mistakenly prioritize spurious features, thereby amplifying spurious bias. In contrast, our method explicitly targets dimensions influenced by spurious bias and re-balances the model's reliance on features, reducing the model's dependency on spurious information.

Furthermore, unlike many variable selection methods that require explicit supervision (e.g., labels or statistical relationships) to mitigate spurious bias, LaSAR operates in an unsupervised setting where group labels indicative of spurious features are unavailable. By leveraging misclassification signals to estimate spuriousness scores, our method is better suited for scenarios where group annotations are costly or infeasible, offering a practical and scalable solution to the challenge of spurious bias mitigation.

| Method | Time complexity |
|---|---|
| JTT (Liu et al., 2021) | $O(NE\tau_{\text{opt}})$ |
| AFR (Qiu et al., 2023) | $O(N_{\text{Ide}}\tau_{\text{fw}} + EN_{\text{ret}}E\tau_{\text{ll}})$ |
| DFR (Kirichenko et al., 2023) | $O(EN_{\text{ret}}E\tau_{\text{ll}})$ |
| LaSAR | $O(E(N_{\text{Ide}}D\tau_{\text{fw}} + N_{\text{ret}}E\tau_{\text{ll}}))$ |

Table 6: Computation complexity comparison with different reweighting methods.

## A.6 DATASET DETAILS

Table 7 gives the details of the two image and two text datasets used in the experiments. Additionally, the ImageNet-9 dataset (Xiao et al., 2021) has 54600 and 2100 training and validation images, respectively. The ImageNet-A (Hendrycks et al., 2021) dataset has 1087 images for evaluation.

## A.7 TRAINING DETAILS

Table 8 and Table 9 give the hyperparameter settings for ERM and LaSAR training, respectively.

## A.8 VISUALIZATIONS ON CORE AND SPURIOUS DIMENSIONS

We provide visualizations on the value distributions of neuron activations for the identified core and spurious dimensions from Fig. 5 to Fig. 8. The spurious and core dimensions selected for visualizations are obtained by first sorting the dimensions based on their spuriousness scores and then selecting three spurious dimensions that have the largest scores and three core dimensions that have the smallest scores. Note that a dimension does not exclusively represent a core or a spurious feature; it represents a mixture of them with both kinds of feature being relevant or irrelevant to the target class based on the training data.

On the CelebA dataset, as shown in Fig. 5, samples that highly activate the core dimensions have both males and females; thus, the core dimensions do not have gender bias. For samples that highly activate the identified spurious dimensions, all of them are females, demonstrating a strong reliance on the gender information. In Fig. 6, samples that highly activate the identified spurious dimensions (right side of Fig. 6) tend to have slightly darker hair colors or backgrounds, as compared with samples that highly activate the identified core dimensions (left side of Fig. 6). With the aid of the heatmaps, we observe that these spurious dimensions mostly represent a person's face, which is irrelevant to the target class.

On the Waterbirds dataset, as shown in Fig. 7, for the landbird class, the identified core dimensions mainly represent certain features of a bird and land backgrounds. For the identified spurious dimen-

| Class | Spurious feature | Train | Val | Test |
|---|---|---|---|---|
| \multicolumn{5}{c}{Waterbirds} | | | | |
| landbird | land | 3498 | 467 | 2225 |
| landbird | water | 184 | 466 | 2225 |
| waterbird | land | 56 | 133 | 642 |
| waterbird | water | 1057 | 133 | 642 |
| \multicolumn{5}{c}{CelebA} | | | | |
| non-blond | female | 71629 | 8535 | 9767 |
| non-blond | male | 66874 | 8276 | 7535 |
| blond | female | 22880 | 2874 | 2480 |
| blond | male | 1387 | 182 | 180 |
| \multicolumn{5}{c}{MultiNLI} | | | | |
| contradiction | no negation | 57498 | 22814 | 34597 |
| contradiction | negation | 11158 | 4634 | 6655 |
| entailment | no negation | 67376 | 26949 | 40496 |
| entailment | negation | 1521 | 613 | 886 |
| neither | no negation | 66630 | 26655 | 39930 |
| neither | negation | 1992 | 797 | 1148 |
| \multicolumn{5}{c}{CivilComments} | | | | |
| neutral | no identity | 148186 | 25159 | 74780 |
| neutral | identity | 90337 | 14966 | 43778 |
| toxic | no identity | 12731 | 2111 | 6455 |
| toxic | identity | 17784 | 2944 | 8769 |

Table 7: Numbers of samples in different groups and different splits of the four datasets.

| Hyperparameters | Waterbirds | CelebA | ImageNet-9 | MultiNLI | CivilComments |
|---|---|---|---|---|---|
| Initial learning rate | 3e-3 | 3e-3 | 1e-3 | 1e-5 | 1e-3 |
| Number of epochs | 100 | 20 | 120 | 10 | 10 |
| Learning rate scheduler | CosineAnnealing | CosineAnnealing | MultiStep[40,60,80] | Linear | Linear |
| Optimizer | SGD | SGD | SGD | AdamW | AdamW |
| Backbone | ResNet50 | ResNet50 | ResNet18 | BERT | BERT |
| Weight decay | 1e-4 | 1e-4 | 1e-4 | 1e-4 | 1e-4 |
| Batch size | 32 | 128 | 128 | 16 | 16 |

Table 8: Hyperparameters for ERM training.

sions, they mainly represent water backgrounds, which are irrelevant to the landbird class based on the training data. For the waterbird class, as shown in Fig. 8, the identified core dimensions mostly represent certain features of a bird and water backgrounds, while the identified spurious dimensions mainly represent land backgrounds.

| Hyperparameters | Waterbirds | CelebA | ImageNet-9 | MultiNLI | CivilComments |
|---|---|---|---|---|---|
| Learning rate | 1e-3 | 1e-3 | 1e-3 | 1e-5 | 1e-3 |
| Number of batches per epoch | 200 | 200 | 200 | 200 | 200 |
| Number of epochs | 40 | 40 | 1 | 60 | 60 |
| Optimizer | SGD | SGD | SGD | AdamW | AdamW |
| Batch size | 128 | 128 | 128 | 128 | 128 |

Table 9: Hyperparameters for LaSAR.

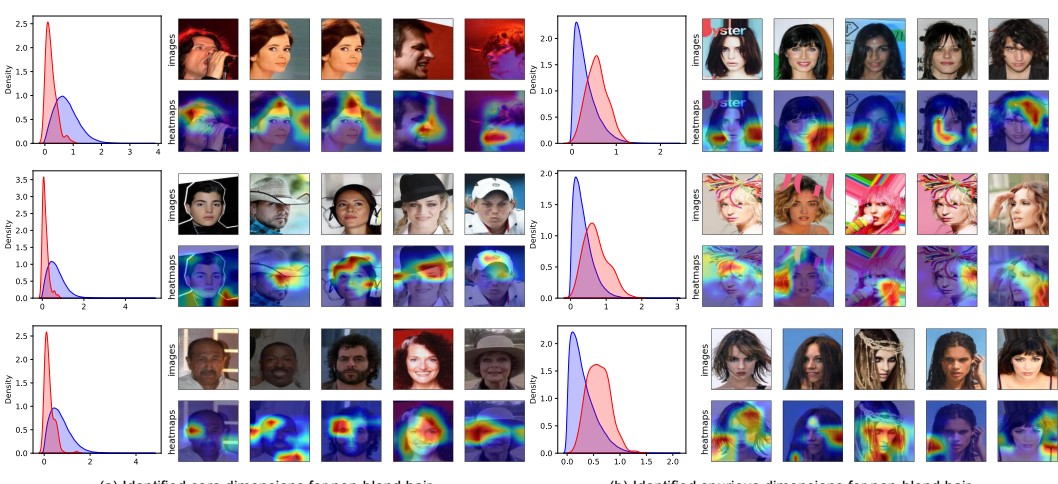

(a) Identified core dimensions for non-blond hair      (b) Identified spurious dimensions for non-blond hair

Figure 5: Value distributions along with representative samples for spurious and core dimensions, respectively, based on the non-blond hair samples in the CelebA dataset.

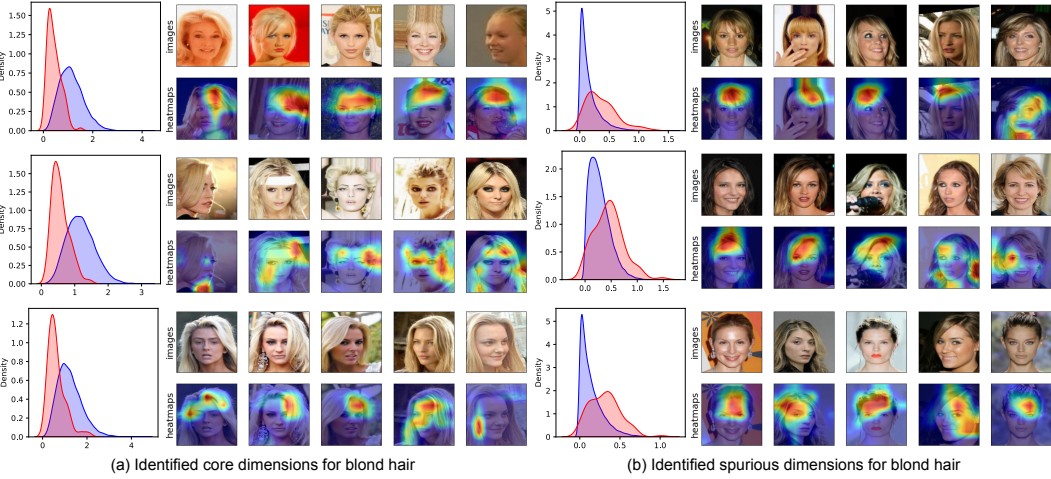

(a) Identified core dimensions for blond hair      (b) Identified spurious dimensions for blond hair

Figure 6: Value distributions along with representative samples for spurious and core dimensions, respectively, based on the non-blond hair samples in the CelebA dataset.

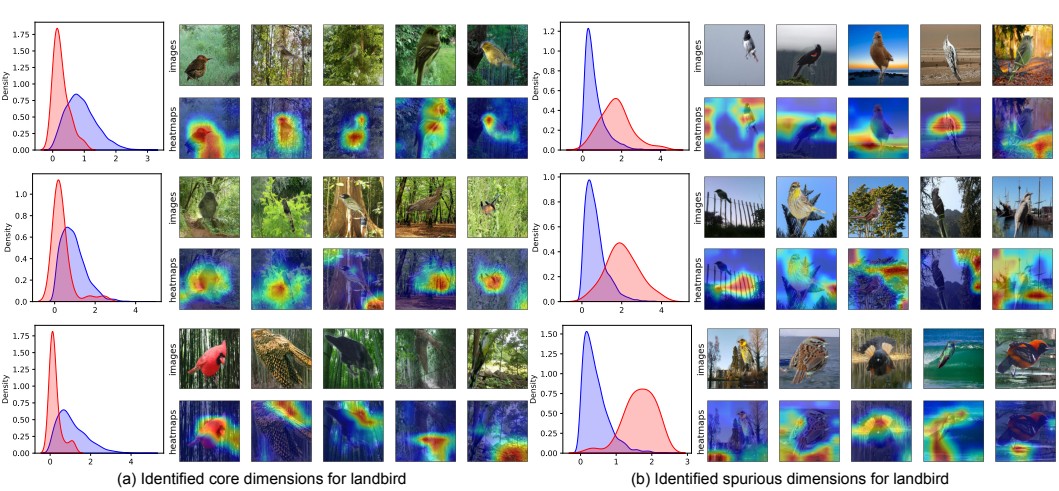

Figure 7: Value distributions along with representative samples for spurious and core dimensions, respectively, based on the landbird samples in the Waterbirds dataset.

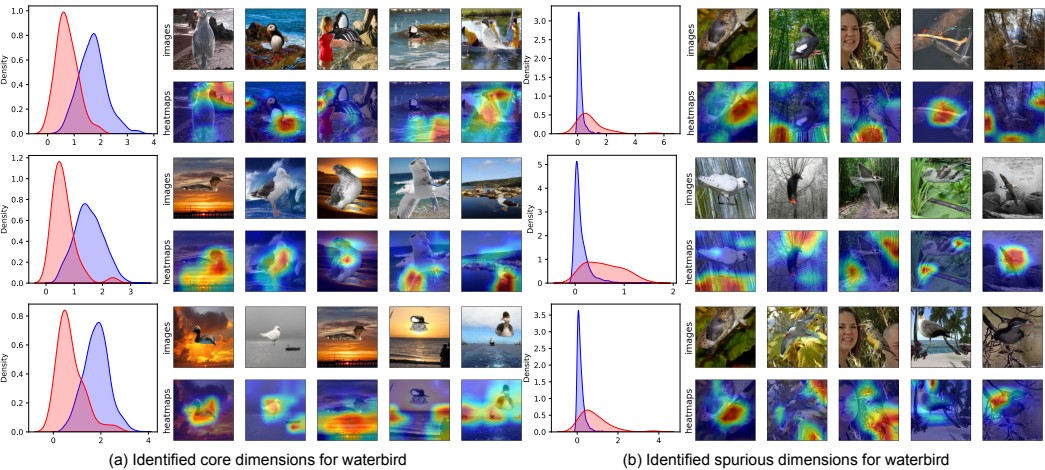

Figure 8: Value distributions along with representative samples for spurious and core dimensions, respectively, based on the waterbird samples in the Waterbirds dataset.

