# OpenReview forum: "Mitigating Spurious Bias with Last-Layer Selective Activation Retraining"
_ICLR.cc/2025/Conference — Submitted to ICLR 2025_

### Official Review · Reviewer_Qpa6 · 2024-11-01

**Soundness:** 2
**Presentation:** 3
**Contribution:** 2
**Rating:** 3
**Confidence:** 3

**Summary:**

The paper proposes a new method for mitigating spurious bias in an unsupervised fashion.

It tries to detect non-essential features based on the pattern of errors, mask them away, and retrain the last layers. The method is compared against other methods in image and text datasets.

**Strengths:**

The paper deals with a relevant problem. It proposes a new method that is, to the best of my knowledge original. And presents an evaluation in relevant benchmarks.

**Weaknesses:**

My main concern is that I find the motivation for the method not very strong. I believe the authors don't provide strong evidence for the main assumptions that motivate the methods.

Particularly, core assumptions for the method are that
1.  some features in the latent embedding that are responsible for encoding the *spurious correlation are confined to single neurons*, and can be masked away. It is a bit unclear to me whether this is true. For instance, maybe some component that is not entirely aligned with any specific neuron could be responsible for encoding this spurious feature.
2. spurious features can be distinguished from core features by looking at the error density. And while the toy example motivates this, It seems the pattern we see in Fig2(b) for spurious vs core features is very different from what we see in Fig 4.
Overall, I think these are two very important assumptions of the method that should be more clearly demonstrated

**Questions:**

Some minor concerns.
- I don't understand, why not provide visualizations using linear dimensionality reduction for the motivating example (section 3.2), since you are using a linear model. Using T-SNE somehow confuses the example
- How were the baselines implemented? are they openly available (it could make sense to provide the links) or did you re-implemented them
- How many spurious features were masked away in each of the examples

---

> ### Author Response · Authors · 2024-11-25
>
> 1. **Regarding the two very important assumptions of the method:
> (1) some features in the latent embedding that are responsible for encoding the spurious correlation are confined to single neurons, and can be masked away, and (2) spurious features can be distinguished from core features by looking at the error density.**
>
>     Thank you for raising this concern. We would like to clarify that our method does not assume that a neuron exclusively represents a spurious feature that can simply be masked. In reality, a neuron may encode a mixture of spurious and core features. Our method does not attempt to distinguish spurious features from core features directly. Instead, it focuses on selecting neurons that produce high activations for an undesired class, highlighting the model's strong reliance on spurious correlations between the undesired class and certain features represented by those neurons.
>
>     To support the motivation behind our proposed method, we provide a comprehensive theoretical analysis in the paper: (1) outlining the principle for selecting neurons (**Proposition 1**), (2) proving that the spuriousness score based on the distribution of neuron activations can effectively identify the desired neurons (**Theorem 1**), and (3) demonstrating that our selective activation retraining method can effectively mitigate spurious bias (**Theorem 2**). For further details, please refer to the revised paper.
>
>
> 2. **Why not provide visualizations using linear dimensionality reduction for the motivating example?**
>
>     Thank you for raising this concern. We chose T-SNE because we used the multi-dimensional input data for visualization and there is no linear relationship between input dimensions.
>
> 3. **How were the baselines implemented? are they openly available (it could make sense to provide the links) or did you re-implemented them**
>
>    The baselines we compared against are openly available, and their respective links are provided in the original papers. To ensure a fair and reliable comparison, we used the reported performance metrics from these methods as presented in their papers, as they were evaluated on the same datasets and under the same settings. Leveraging the authors' official results avoids potential discrepancies that could arise from re-implementations, ensuring the integrity of our evaluation. We believe this approach provides the most accurate and convincing comparison between our method and the baselines.
>
> 4. **How many spurious features were masked away in each of the examples?**
>
>    Thank you for the question. Indeed, our method masks the same last-layer neurons for all examples. The table below presents the number of masked neurons and their corresponding percentages among all neurons across the four datasets.
>
> | waterbirds | CelebA | MultiNLI | CivilComments |
> |------------|--------|----------|---------------|
> | 1922       | 1617   | 701      | 632           |
> | 93.8%      | 80.0%  | 91.3%    | 82.3%         |

---

> > ### Comment · Reviewer_Qpa6 · 2024-11-28
> >
> > I read the answer from the author. The answer did not address my concerns and did not change my assessment . I keep my overall score

---

> > > ### Author Response · Authors · 2024-11-28
> > >
> > > Thank you for taking the time to review our responses and share your thoughts. We apologize for not adequately addressing your concerns in our previous explanation.
> > > We greatly value your feedback and would appreciate the opportunity to further clarify and refine our responses during the discussion period.
> > >
> > > Since your feedback did not specify which aspects of our response were unsatisfactory, we kindly request further clarification on the points you found insufficient. This will allow us to provide a more precise explanation. In the meantime, we would like to revisit your main concerns about the two assumptions and provide additional clarifications:
> > >
> > > - **Regarding Assumption 1: spurious features confined to single neurons**
> > >
> > >     We agree that it may *not* hold true that some features in the latent embedding that are responsible for encoding the spurious correlation are confined to single neurons. In fact, these features may spread to multiple neurons, as you may observe, for example, in Fig. 5(b), where all the dimensions are predominately activated on images of females. Similarly, core features may also be represented across multiple neurons. In this realistic setting, our method identifies neurons affected by spurious bias, which we define as those with high activations for undesired classes or neurons highly activated during incorrect predictions. To reflect this and improve clarity, we have refined our definition on spurious and core dimensions in the revised manuscript (Line 238-242). Furthermore, we provided theoretical evidence in Section 3.3 to demonstrate the effectiveness of our approach under these general conditions.
> > >
> > > - **Regarding Assumption 2: error density can distinguish spurious and core features**
> > >
> > >     We acknowledge your observation that the patterns for spurious versus core features differ between synthetic experiments and real-world settings. While this discrepancy exists, it does not diminish the effectiveness of our method. The reason is that our approach relies on the median value of the distributions (as described in Eq. (5)) to identify spurious dimensions. Therefore, our method is relatively robust to the variations in the distributions compared with using the whole distribution patterns.  Furthermore, the theoretical evidence we provided in Section 3.3 demonstrates that our approach can effectively identify spurious dimensions, regardless of potential distribution variations.
> > >
> > > We appreciate the opportunity to address your concerns further during the discussion period. If there are additional aspects that remain unclear or unsatisfactory, please let us know, and we will provide a more tailored explanation. Thank you again for your thoughtful review and for contributing to the improvement of our work.

---

### Official Review · Reviewer_fuS4 · 2024-11-01

**Soundness:** 3
**Presentation:** 3
**Contribution:** 3
**Rating:** 6
**Confidence:** 4

**Summary:**

In this work, authors propose a debiasing method that works by retraining only the last layer (classification layer) in order to re-weight different factors in the latent representation. The assumption is that latent factors carry different information (source, spurious, noise) which can be effectively filtered out from the classification layer by reweighting. They test their method on standard debiasing benchmarks such as celeba, waterbirds, multinli and civil comments.

**Strengths:**

- The paper tackles a very important issue, which is learning unbiased models from biased data.
- The proposed method does not need any kind of annotation on the bias, and it just leverages the class label (unsupervised debiasing)
- The reported results show improvement w.r.t other methods

**Weaknesses:**

Here are my main concerns about the work:

- The authors' assumption is that latent representation can be factorized in source, spurious and noise components. This is clearly shown in the toy example; however it is not clear why this should also happen in representations extracted from deep neural networks on complex data. It might not be so simple to factor out single components in the learned representations, as they might be intertwined and correlated. Can you provide some more theoretical backing of this method?

- I think that validation on more difficult datasets such as 9-Class ImageNet / ImageNet-A (https://openreview.net/forum?id=2OqZZAqxnn) should be added to the experimental validation.

- The related work section should be updated a bit with relevant works in the area (e.g. [1-6])

[1] Bahng, Hyojin, et al. "Learning de-biased representations with biased representations." International Conference on Machine Learning. PMLR, 2020.

[2] Tartaglione, Enzo, et al. "End: Entangling and disentangling deep representations for bias correction." Proceedings of the IEEE/CVF conference on computer vision and pattern recognition. 2021.

[3] Y.-K. Zhang, Q.-W. Wang, D.-C. Zhan, and H.-J. Ye, “Learning debiased representations via conditional attribute interpolation” in Proceedings of the IEEE/CVF Conference on Computer Vision and Pattern Recognition, 2023.

[4] Barbano, Carlo Alberto, et al. "Unbiased Supervised Contrastive Learning." ICLR. 2023.

[5] Zhang, Yi, et al. "Poisoning for Debiasing: Fair Recognition via Eliminating Bias Uncovered in Data Poisoning." ACM Multimedia 2024. 2024.

[6] Wang, Yining, et al. "Navigate Beyond Shortcuts: Debiased Learning through the Lens of Neural Collapse." Proceedings of the IEEE/CVF Conference on Computer Vision and Pattern Recognition. 2024.

**Questions:**

- Could also provide results in terms of balanced accuracy (other than accuracy and WGA)?
- I think that retraining on a part of the validation set may lead to "unfair" comparison with baseline methods that are not trained also on validation data
- Reweighting the classification layer essentially does not "remove" the bias from the whole model, but it is just a correction. Do you think this might be an issue in certain cases?
- I do not see a clear difference between core activation maps and spurious activations maps for CelebA in Fig. 4., the spurious heatmaps even seem a bit more focused on the hair (which is the target task).
- Using your method do you think it would be possible to provide pseudo-labels for the training data in order to use a supervised debiasing method?

---

> ### Author Response · Authors · 2024-11-25
>
> 1. **Regarding learned latent representations cannot be factored into source, spurious and noise components, and theoretical backing of this method**
>
>     Thanks for raising this concern. We apologize for the confusion. Actually, we do not assume that learned representations can be factored into individual components, and our method does not aim to disentangle spurious dimensions from core dimensions. In a nutshell, we aim to identify dimensions that reflect a model's spurious prediction behavior and then block the contributions from these dimensions for retraining. To better explain our method, we have provided a theoretical analysis in Section 3.3. Specifically, for dimension (neuron) selection, our principal is to select those that reflect spurious prediction behaviors of the model, as formalized in **Proposition 1**. We call these dimensions as spurious dimensions. We have refined our definition on the spurious and core dimensions in the revised paper (Line 233-236). To further justify our algorithmic designs, we provide two theorems in Section 3.3. **Theorem 1** proves that the spuriousness score defined in Eq. (5) indeed selects dimensions that meet our selection principal. **Theorem 2** proves that our method, LaSAR, can effectively mitigate spurious bias in the model. Please kindly refer to our revised paper for details.
>
>
> 2. **Regarding experiments on 9-Class ImageNet / ImageNet-A**
>
>    Thanks for the suggestion. We have provided the experimental results on the ImageNet-9 and ImageNet-A datasets in Table 7 in the revised paper. Our method still works in this challenging setting.
>
> 3. **The related work section should be updated a bit with relevant works in the area**
>
>     Thanks for the suggestion. We have added discussions on the given works in the related work section. Please kindly refer to our revised paper for details.
>
>
> 4. **Could also provide results in terms of balanced accuracy (other than accuracy and WGA)?**
>
>     Thanks for the suggestion. We have provided balanced accuracy for our methods. We hope to emphasize that while balanced accuracy may provide an overall measure of the performance of a method, WGA is the most commonly used metric in this area and the majority of works reported performance under this metric. Other than WGA, we also provied Acc. gap in our paper, which measures the performance discrepancy across different data groups. We hope these metrics help reinforce your confidence in the effectiveness of our methods.
>
>     | Method          | Waterbirds       | CelebA           | MultiNLI         | CivilComments    |
>     |-----------------|------------------|------------------|------------------|------------------|
>     | LaSAR           | $94.0_{\pm 0.2}$ | $89.1_{\pm 1.1}$ | $80.8_{\pm 0.5}$ | $85.9_{\pm 0.0}$ |
>     | LaSAR$^\dagger$ | $93.7_{\pm 0.2}$ | $89.7_{\pm 0.3}$ | $80.0_{\pm 0.4}$ | $86.2_{\pm 0.1}$ |
>
> 5. **I think that retraining on a part of the validation set may lead to "unfair" comparison with baseline methods that are not trained also on validation data**
>
>     Thanks for raising this concern. We have specifically marked methods (with $^\dagger$) that use validation set for training in Table 1 and Table 2, allowing for fair comparison.
>
>
> 6. **Reweighting the classification layer essentially does not "remove" the bias from the whole model, but it is just a correction. Do you think this might be an issue in certain cases?**
>
>     Thanks for your question. Our **Theorem 2** proves that this in generally would not be an issue, and our method can bring the retrained model very closer to the unbiased model in the parametric space than its original version. If the retrained model does not meet certain robustness criteria in practice, then retraining the whole model would be necessary. However, as shown in the **Corollary 1** in Appendix, the unbiased model can only be achieved if the whole model can be retrained with unbiased data. Nevertheless, it is worth to mention that our method is an efficient and post-hoc bias mitigation method which may be of independent interest.

---

> > ### Author Response · Authors · 2024-11-25
> >
> > 7. **I do not see a clear difference between core activation maps and spurious activations maps for CelebA in Fig. 4., the spurious heatmaps even seem a bit more focused on the hair (which is the target task).**
> >
> >     Thank you for raising this concern. First, we want to clarify that neurons do not exclusively represent either spurious or core features; instead, they typically encode a mixture of both. Consequently, activation maps may sometimes highlight mixed features. To address this, we have refined our definition of a spurious dimension as one that exhibits high activation values for an undesired class (Lines 233–236). For example, the dimension shown in Fig. 4(b) is considered spurious under this definition because it activates land backgrounds for the waterbird class, even though these features are beneficial for recognizing the landbird class, as evidenced by the high prediction error.
> >
> >     Regarding the CelebA dataset, the highlighted regions in Fig. 4(d) primarily cover people’s faces and appear relatively darker than those in Fig. 4(c). These features are beneficial for predicting the undesired non-blond hair class, as indicated by the high prediction error for the blond hair class shown in Fig. 4(d). To further illustrate the concept of spurious and core dimensions, we provide additional visualizations in Figs. 6 through 9.
> >
> >
> > 8. **Using your method do you think it would be possible to provide pseudo-labels for the training data in order to use a supervised debiasing method?**
> >
> >     Thank you for the question. We believe it is possible to generate pseudo-labels for the training data using our method. For instance, spuriousness scores for each dimension of the latent representations could be used to distinguish samples with different spurious features. While this is beyond the scope of the current paper, it is a promising direction for future research.

---

> > > ### Comment · Reviewer_fuS4 · 2024-11-25
> > >
> > > I thank the authors for the thorough response and the additional experiments.
> > >
> > > I think that the new presentation about my first concern (factorized components) is now more suited for the proposed method.
> > >
> > > Regarding the new results on ImageNet, I see there is a large gap between the performance achieved by LaSaR and the other baselines (especially on ImageNet-A). Is the setup used comparable to the reported baselines? How do you justify this? The gap in the other experiments does not seem so high.
> > >
> > > (Minor) Also, I am still doubtful about the activation maps. In my opinion, they are inconclusive for too many examples

---

> > > > ### Author Response · Authors · 2024-11-26
> > > >
> > > > Thank you for your thoughtful feedback and for acknowledging the improvements in our response and additional experiments.
> > > >
> > > > Regarding the performance gap on ImageNet-A, thank you for highlighting this concern. The gap can be attributed to the use of pre-trained ImageNet weights in LaSAR, which substantially boosts performance compared to other baselines. To address this and ensure a fair comparison, we have included new results (Table 7 in the revised paper) where the model is trained from scratch. These updated results show a more reasonable gap, consistent with the performance observed in other experiments.
> > > >
> > > > With respect to the activation maps, we acknowledge that some examples may appear inconclusive due to neurons representing mixed features and certain activation maps not displaying clear patterns. However, providing more examples allows for the observation of global trends. For instance, in the CelebA dataset, for the non-blond hair class, images that strongly activate the identified core dimensions include both males and females (Fig. 6(a)), whereas images activating the spurious dimensions are predominantly females with bright backgrounds (Fig. 6(b)). Similarly, for the blond hair class, images activating the spurious dimensions tend to have darker colors (Fig. 7(b)) compared to those activating the core dimensions (Fig. 7(a)). These patterns illustrate the global consistency of the activation maps in demonstrating core and spurious dimensions.
> > > >
> > > > We appreciate your constructive comments and hope that these updates address your concerns.

---

> > > > > ### Comment · Reviewer_fuS4 · 2024-11-26
> > > > >
> > > > > Thank you for the response and the additional experiments. I see now the performance is more reasonable and in line with previous experiments.
> > > > > About the activation maps, personally, I found them not very informative (in general) when there is not a consistent and clear difference. I suggest authors move them to the appendix and prioritize experiments on ImageNet in the main text.
> > > > >
> > > > > Overall authors have addressed my concerns and I am satisfied with the rebuttal. I will partially increase my score, although I think that the paper needs more polishing and overall improvements (e.g. results of some experiments).
> > > > >
> > > > > For future iterations, I also suggest authors show a comparison in terms of speed and complexity of the different methods, I think LaSAR can have a competitive advantage in this setting.

---

> > > > > > ### Author Response · Authors · 2024-11-27
> > > > > >
> > > > > > Thank you for your thoughtful feedback and suggestions. We are pleased to hear that our response and additional experiments addressed your concerns, and we appreciate your acknowledgment of the improvements in our work.
> > > > > >
> > > > > > Regarding the activation maps, we understand your perspective and have moved these visualizations to the appendix and moved the ImageNet experiment to the experiment section in the revised paper.
> > > > > >
> > > > > > We also value your suggestion to include comparisons of speed and complexity across methods. This is an excellent point. We have incorporated such analyses in Section A.4 in Appendix to highlight this aspect.
> > > > > >
> > > > > > Finally, thank you for your time and insights, which have been invaluable in enhancing the quality of this research.

---

### Official Review · Reviewer_Ymhg · 2024-11-04

**Soundness:** 2
**Presentation:** 2
**Contribution:** 2
**Rating:** 6
**Confidence:** 2

**Summary:**

In this paper, the authors propose a novel method to self-identify spurious features and mitigate the spurious bias by retraining the last classification layer. In general, the idea of using neuron activations before the last classification layer, coupled with their final prediction outcomes, to provide self-identifying information on whether the neurons represent spurious features seems interesting.

**Strengths:**

The authors did extensive experiments.

**Weaknesses:**

The writing in some places is unclear, in particular, they did not clearly explain the behind reasoning of the proposed method to identify the spurious features. The did not use some theoretical results to support the proposed method.

**Questions:**

1, why did you define the spuriousness score as (5)? To help readers understand the behind rationale, I think the authors may need to add more explanation.

2, In line 157, the authors mentioned that WGA is the accuracy on the worst performing data group in the test set $\mathcal{D}_{test}$. However, they used argmax in the formula of WGA, it seems problematic since argmax will output a group label rather than the value of accuracy and the argmax will output the best performing data group in terms of accuracy rather than the worst performing data group.

3, In Section 3.2, they used a synthetic motivating example. It may be better to use a real motivating example.

4, In line 321, the authors may want to say "equation (6) and equation (7)" rather than equation 6 and equation 7.

5, I think the study objective in this paper is quite similar to variable selection in statistics. We can use many penalties such as L1 penalty to remove those spurious features. I do not see the advantages of the proposed method compared with those variable selection methods in statistics. The authors may need to discuss this point.

---

> ### Author Response · Authors · 2024-11-25
>
> 1. **Regarding the reasoning of the proposed method to identify the spurious features and theoretical results**
>
>    Thanks for raising this concern. We have added a theoretical analysis in the paper to offer insights into our detection method. In summary, we first gave  in **Proposition 1** the principal for selective activation, which is the core of LaSAR. Proposition 1 specifies the property of a neuron that is affected by spurious bias. Specifically, our selection method select neurons that have high activation values but not beneficial for predicting the target. Then, we theoretically showed in **Theorem 1** that our proposed spuriousness score indeed selects neurons in the last layer in a way that follows the proposed principal in Proposition 1. In other words, the score in general can effectively identify neurons affected by spurious bias. Finally, we proved in **Theorem 2** that LaSAR can mitigate spurious bias in a model by bringing the model closer to the unbiased one in the parameter space. Please kindly refer to Section 3.3 in the revised paper. All changes are marked in blue color.
>
> 2. **Why did you define the spuriousness score as (5)?**
>
>     Thanks for the suggestion. The score is defined such that a large difference between $\mu_{\text{mis}}$ and $\mu_{\text{cor}}$, i.e., a large $\delta_{i}^y$, indicates a high likelihood of the $i$'th dimension being affected by the spurious bias in the model. In other words, the model incorrectly amplifies a spurious feature in the neuron activation when it should not. In contrast, a negative $\delta_{i}^y$ shows the importance of the $i$'th dimension for predictions as most correctly predicted samples tend to have high activation values on this dimension, while most incorrectly predicted samples have low activation values. We have added this explanation in the revised paper.
>
> 3. **Regarding the definition of WGA in line 157**
>
>     Thanks for pointing this out. We have corrected this in the revised paper.
>
>
> 4. **In Section 3.2, they used a synthetic motivating example. It may be better to use a real motivating example.**
>
>     Thanks for the suggestion! To better highlight our motivation and contribution, we have replaced the synthetic experiment with a theoretical analysis section and moved the synthetic experiment to the experiment section. Please kindly refer to the revised paper for more details.
>
>
> 5. **In line 321, the authors may want to say "equation (6) and equation (7)" rather than equation 6 and equation 7.**
>
>    Thanks for pointing this out. We have corrected this in the revised paper.
>
> 6.  **Regarding relation to variable selection in statistics**
>
>     Thanks for providing this new perspective. Variable selection methods, such as L1 regularization, does not take into account whether those variables represent a spurious feature or not, and they can be easily biased by the imbalanced distribution of the data. In fact, using traditional variable selection methods may actually amplify spurious bias as they are based on statistics of the data.  In contrast, our method not only exploits the distributions of neuron activation values but also utilizes the label information to jointly characterize the properties of spurious features and incorporate them into our algorithmic design.

---

### Official Review · Reviewer_pwXU · 2024-11-04

**Soundness:** 2
**Presentation:** 3
**Contribution:** 1
**Rating:** 3
**Confidence:** 4

**Summary:**

This paper introduces Last-layer Selective Activation Retraining (LaSAR), which identifies and mitigates spurious bias without requiring external supervision or group labels. The key point lies in observing that neuron activations combined with prediction outcomes can self-identify spurious features, and then using this information to selectively block spurious neurons during last-layer retraining. The method works as a practical post-hoc tool in standard ERM training settings, and requires no additional annotations beyond class labels. Authors compare their method with competitive baselines such as JTT, and DFR and show some improvement in worst group accuracy on a benchmark with 4 datasets.

**Strengths:**

Key Strengths:

1. Spurious neuron identification: The proposed LaSAR framework introduces an interesting approach to identify spurious neurons using activation patterns and prediction outcomes, providing a self-guided mechanism for bias detection.

2. Practical Utility: The method works as a post-hoc tool in standard ERM training settings, making it highly practical for real-world applications.

**Weaknesses:**

1. The contribution of this paper is severely limited. Indeed, the core intuition that using (i) misclassified examples of validation data, (ii) and retraining all layers or the linear head to reduce reliance on spurious features has been demonstrated previously with methods such as JTT and AFR. How is LaSAR fundamentally different from AFR?

2. Lack of fair comparison. Although JTT and AFR need group information on the validation data only to tune hyper-parameters. They can be tuned using the worst-class accuracy. Authors should therefore compare their method with JTT and AFR when tuned on worst-class accuracy.

3. No theoretical guarantees are provided about the convergence and stability of the selective activation retraining process, even on synthetic data.

**Questions:**

1. How does the method ensure that it doesn't accidentally block neurons representing valid but complex feature combinations rather than truly spurious correlations?
2.  How does the method handle cases where features might be spurious in some contexts but valid in others?
3. (Also related to 1.) There has been evidence that neurons may learn polysemantic features. What is the impact of LaSAR in case neurons may learn linear combinations of spurious and core features?

---

> ### Author Response · Authors · 2024-11-25
>
> 1. **Regarding the contribution**
>
>    We hope to emphasize that our contribution does not focus on "misclassified examples of validation data and retraining all layers or the linear head". Indeed, we focus on mitigating spurious bias from the perspective of neuron masking, which has the potential to train a robust model **without any group information**. Our contributions include: (1) an automatic detection framework that identifies neurons affected by spurious bias, (2) a spurious bias mitigation method that blocks spurious neurons for retraining, and (3) we demonstrate the effectiveness of our proposed method on benchmark datasets and provide theoretical analyses (in the revised paper) to justify our design choices.
>
> 2. **How is LaSAR fundamentally different from AFR?**
>
>     Thanks for the question. AFR is a sample-level method in the sense that it mitigates spurious bias by up-weighting samples that have high prediction losses. In contrast, LaSAR works in the latent embedding space and mitigates spurious bias by selectively blocking neurons that are identified as spurious. Moreover, AFR identifies which samples are important for retraining using the prediction loss from an ERM trained model. LaSAR blocks certain neurons for all the training samples and does not prefer one sample over another during retraining. LaSAR utilizes the distributions of neuron activations to identify spurious bias. Moreover, LaSAR can work in the completely unsupervised spurious bias mitigation setting.
>
>
> 3. **Regarding comparison with JTT and AFR when tuned on worst-class accuracy.**
>
>      Thank you for bringing this up. In the table below, we provide a comparison between AFR, JTT, and LaSAR, where JTT and AFR were tuned using worst-class accuracy. Typically, models tuned with worst-class accuracy tend to perform worse than those tuned with worst-group accuracy, as the table demonstrates. We want to emphasize that, even without access to group information, LaSAR is already competitive with state-of-the-art methods tuned using worst-group accuracy.
>
>     | Method | Waterbirds | CelebA |
>     |--------|------------|--------|
>     | JTT    | 84.2$\_{\pm 0.5}$       | 52.3$\_{\pm 1.8}$   |
>     | AFR    | 89.0$\_{\pm 2.6}$       | 68.7$\_{\pm 1.7}$   |
>     | LaSAR  | 91.8$\_{\pm 0.8}$      | 83.0$\_{\pm 2.8}$   |
>
>
> 4. **Regarding theoretical guarantees about the convergence and stability of the selective activation retraining process**
>
>     Thanks for pointing this out. In the revised paper, we have provided a theoretical analysis on a linear model with synthetic data. The main results we obtained are: (1) the spuriousness score indeed selects neurons that are affected by spurious bias (**Theorem 1**), and (2) the selective activation retraining process will converge to a point in the parametric space of the model closer to the unbiased model (**Theorem 2**). Please kindly refer the Section 3.3 in the revised paper for details.
>
> 5. **How does the method ensure that it doesn't accidentally block neurons representing valid but complex feature combinations rather than truly spurious correlations?**
>
>     Thank you for raising this important point. We apologize for any confusion and would like to clarify that our method does not explicitly aim to block purely spurious features or activate purely core features. As shown in Eq. (11), even for a simple model, neuron activation represents a mixture of spurious and core components from the input. The principle behind our selective activation method is explained in **Proposition 1**. Fundamentally, we aim to identify spurious behaviors in the model by analyzing neuron activations. In our theoretical framework, it is $\gamma^T\mathbf{w}_{\text{spu},i}<0$. Blocking the identified neurons effectively removes contributions arising from these spurious behaviors. While this approach cannot fully ensure that it won’t unintentionally block neurons representing valid but complex feature combinations, both the theoretical result (**Theorem 2**) and empirical evidence (performance on benchmark datasets) indicate that the tradeoff is worthwhile.
>
>
> 6. **How does the method handle cases where features might be spurious in some contexts but valid in others?**
>
>     Thank you for raising this question. As mentioned in Lines 229–232, we argue that in a well-defined classification task, a spurious dimension identified for one class cannot serve as a key contributor to predicting another class. For instance, in a task that involves classifying between "rectangle" and "blue color", a dimension strongly associated with "blue color" for the "rectangle" class cannot reliably predict the "blue color" class when given a blue rectangle, as this would result in ambiguity. Therefore, we block the identified spurious dimensions across all classes.

---

> > ### Author Response · Authors · 2024-11-25
> >
> > 7. **There has been evidence that neurons may learn polysemantic features. What is the impact of LaSAR in case neurons may learn linear combinations of spurious and core features?**
> >
> >     Thanks for the insightful question. Our theoretical analysis indeed analyzes this case, as you may observe the model we adopted in Eq. (11). In such a setting, as shown by Theorem 2, LaSAR can still bring the retrained model closer to the unbiased one in the parametric space. Due to the inherent mechanism of LaSAR, it cannot optimize the weights for the spurious components, but it can optimize the weights for the core components.

---

> > > ### Comment · Reviewer_pwXU · 2024-11-26
> > >
> > > I thank the authors for their detailed feedback.
> > >
> > > However, I remain unconvinced about how LaSAR fundamentally differs from AFR. My concerns are as follows:
> > >
> > > 1. LaSAR utilizes an identification dataset (specifically, half of the validation set, as described in Section 4.6). This setup appears similar to AFR.
> > >
> > > 2. The spuriousness score introduced by LaSAR heavily depends on misclassified instances, which closely resembles the approaches taken by AFR or JTT.
> > >
> > > 3. LaSAR also involves retraining the classifier head using half of the validation set, a step that again aligns with AFR.
> > >
> > > While I acknowledge that LaSAR introduces a new perspective by masking potential spurious features using the spuriousness score, it remains unclear why AFR would not inherently achieve a similar outcome through its retraining process.
> > >
> > > Additionally, I appreciate the authors' effort to provide new theoretical motivation. However, the proposed theoretical framework seems somewhat disconnected from LaSAR's practical implementation. Specifically, the theoretical setting assumes a linear classifier, whereas LaSAR operates by identifying spurious features in the feature or representational space.
> > >
> > > Finally, I thank the authors for providing a comparison with JTT and AFR when tuned on worst-class accuracy. I think these experiments should be added to the main paper.

---

> > > > ### Author Response · Authors · 2024-11-27
> > > >
> > > > Thank you for your thoughtful comments and for taking the time to review our work in detail. We appreciate the opportunity to address your concerns and provide further clarification. Below, we respond to each point raised in your feedback:
> > > >
> > > > - **How LaSAR fundamentally differs from AFR and why AFR would not inherently achieve a similar outcome through its retraining process?**
> > > >
> > > >     While our method may appear similar to AFR at first glance, we emphasize that it is fundamentally different in how spurious bias is mitigated. To clarify this distinction, we have added a detailed discussion in **Lines 344–346, along with Lemma 3 and Section A.3 in Appendix**. Here, we provide a brief summary for your reference.
> > > >
> > > >     AFR is fundamentally a **sample-level** method and adjusts the weights of the last layer **indirectly** via retraining with samples with varied importance weights. Our method **directly** forces the weights associated with spurious bias to zero, while refining the remaining weights with retraining.
> > > >
> > > >     The reason why AFR would not inherently achieve a similar outcome can be explained more formally in our theoretical analysis framework.
> > > >     First, let's take a closer look at the training loss in Eq. (22), based on Eq. (25) and Eq. (28), we can express it as the sum of following terms for brevity,
> > > >
> > > >     $$\ell\_{tr}(\mathbf{W},\mathbf{b})=\frac{1}{2}p\mathbb{E}[\psi\_1(\mathbf{u}\_{\text{core}},\mathbf{u}\_{\text{spu}})]+\frac{1}{2}(1-p)\mathbb{E}[\psi\_2(\mathbf{u}\_{\text{core}},\mathbf{u}\_{\text{spu}})]+\frac{1}{2}\psi\_3(\mathbf{u}\_{\text{spu}})$$
> > > >
> > > >     where $p$ is the data generation parameter and is fixed, and $\psi\_1$, $\psi\_2$, and $\psi\_3$ denote functions and their specific forms can be inferred from Eq. (25) and Eq. (28). For last-layer retraining methods in general, the optimal solution for $\mathbf{u}\_{\text{spu}}$ is $\mathbf{u}\_{\text{spu}}^*$ defined in Eq. (18).
> > > >     AFR changes the distribution within the first two expectation terms and jointly updates $\mathbf{u}\_{\text{core}}$ and $\mathbf{u}\_{\text{spu}}$, while there is no optimality guarantee for $\mathbf{u}\_{\text{spu}}$ (i.e., $\psi\_3(\mathbf{u}_{\text{spu}})$ is not considered in AFR).
> > > >     By contrast, LaSAR first ensures that $\mathbf{u}\_{\text{spu}}$ is optimal (Eq. (37)), then by Theorem 2, it moves $\mathbf{u}\_{\text{core}}$ close the the unbiased solution.
> > > >
> > > > - **The proposed theoretical framework seems somewhat disconnected from LaSAR's practical implementation**
> > > >
> > > >     Thank you for raising this concern. We believe our theoretical framework is both analytically feasible and representative of the core principles of LaSAR. As shown in Eq. (11), we use $\mathbf{W}$ to represent the feature extractor and $\mathbf{b}$ to denote the last layer. In our theoretical analysis, $\mathbf{W}$ is fixed, and $\mathbf{b}$ is allowed to change, which closely mirrors how LaSAR is implemented in practice. Specifically, masking the identified neurons across all classes is equivalent to manipulating the weights of the last layer. Furthermore, the basic assumptions in our framework are consistent with the standard approach in the literature [1][2] for analyzing spurious bias.
> > > >
> > > >     [1] Arjovsky et al., Invariant risk minimization, arXiv, 2019.\
> > > >     [2] Ye et al., Freeze then train: Towards provable representation learning under spurious correlations and feature noise, AISTATS, 2023.
> > > >
> > > > - **Adding the new results to the main paper**
> > > >
> > > >     Thanks for your suggestion. We have updated and added the new results in the revised paper.

---

> > > > > ### Comment · Reviewer_pwXU · 2024-11-28
> > > > >
> > > > > Regarding the novelty and differences with AFR, I acknowledge that there may be distinctions in how AFR and LaSAR operate. However, the authors primarily highlight minor differences while ignoring the significant similarities between the two approaches. As outlined in my three points, there is a substantial conceptual overlap. I would like the authors to also highlight these similarities in detail before emphasizing the differences. At this stage, I still believe the similarities far outweigh the dissimilarities.
> > > > >
> > > > > Additionally, thank you for clarifying the notations. That said, I find it unconventional to represent embeddings with W instead of
> > > > > F, even if this notation has been adopted from prior work.
> > > > >
> > > > > I recommend that the authors discuss these points in the introduction or the related work of the next version of the paper.

---

> > > > > > ### Author Response · Authors · 2024-11-29
> > > > > >
> > > > > > Thank you for your thoughtful comments and for acknowledging that there are distinctions between AFR and LaSAR. We appreciate your feedback and the opportunity to clarify further. Below, we address your concern about the conceptual overlap and highlight both the similarities and the distinctions in more detail.
> > > > > >
> > > > > > **Similarities between LaSAR and AFR**
> > > > > >
> > > > > >    - **Use of an identification dataset**:\
> > > > > >     Both LaSAR and AFR leverage an identification dataset to guide the mitigation process. In LaSAR, the set is used to identify spurious dimensions, while AFR uses the set to compute sample-level reweighting for retraining.
> > > > > >
> > > > > > - **Dependence on misclassified instances:**
> > > > > >
> > > > > >     Both methods use signals from misclassified instances as part of their mitigation strategy. In LaSAR, this signal is used to compute spuriousness scores for latent dimensions, whereas AFR uses it to compute sample losses for reweighting.
> > > > > >
> > > > > > - **Retraining the classifier head:**\
> > > > > >     Both approaches involve retraining the classifier head as part of their pipeline. In LaSAR, the spurious dimensions are masked, and the remaining weights are retrained, while AFR modifies sample weights and retrains using reweighted data.
> > > > > >
> > > > > > **Key differences between LaSAR and AFR**
> > > > > >
> > > > > > While the above similarities exist, LaSAR introduces a distinct mechanism and perspective that fundamentally differentiates it from AFR:
> > > > > >
> > > > > > - **Dimensional-level versus sample-level focus:**\
> > > > > >     LaSAR operates at the dimensional level by identifying spurious dimensions in the latent embedding space and directly masking them to mitigate spurious bias. In contrast, AFR functions at the sample level, modifying the weights of individual training samples during the retraining phase. This dimensional-level intervention allows LaSAR to directly address spurious bias at its source in the feature space.
> > > > > >
> > > > > > - **Spuriousness score and feature masking:**\
> > > > > >     LaSAR introduces a spuriousness score to quantify the degree to which each dimension is affected by spurious bias. Based on this score, LaSAR explicitly masks dimensions identified as spurious, ensuring that the model does not rely on these dimensions during retraining. AFR does not perform such explicit masking and instead relies on an indirect reweighting mechanism.
> > > > > >
> > > > > > - **Unsupervised nature of LaSAR:**\
> > > > > >     LaSAR is designed to operate without any group annotations.  Its focus on feature-level spuriousness makes it inherently more suitable for unsupervised spurious bias mitigation. AFR, being sample-focused, does not inherently differentiate varied degrees of spuriousness within the feature space.
> > > > > >
> > > > > > **Why the differences matter**: We recognize that the similarities may suggest conceptual overlap; however, the key innovations of LaSAR lie in its approach to identifying and mitigating spurious bias. By targeting spurious dimensions explicitly, LaSAR provides a novel way to address spurious bias that goes beyond sample-level strategies. Furthermore, our theoretical framework provides additional insights into how LaSAR achieves this, which we believe distinguishes it from AFR.
> > > > > >
> > > > > > We hope this clarification addresses your concerns and highlights the nuanced relationship between the two methods. Should you have further suggestions, we would be happy to refine our manuscript to ensure the distinctions and similarities are both adequately addressed.

---

> > > > > > > ### Comment · Reviewer_pwXU · 2024-12-03
> > > > > > >
> > > > > > > Thank you for your answers. I encourage the authors to include all these changes and discussions in the future version of the paper. I keep my score as I think the paper needs changes to clarify the contribution.

---

> ### Author Response · Authors · 2024-12-03
>
> Thank you for your thoughtful feedback and for reviewing our responses. We sincerely appreciate your encouragement to incorporate the suggested changes and discussions into future iterations of the paper.
>
> We would like to highlight that many of these clarifications and improvements have already been incorporated into the current revision:
>
> - Section 3.3 now provides formal guarantees on LaSAR’s spurious bias mitigation.
> - We clarified the definitions of spurious and core dimensions (L237–238) and explicitly discussed the assumption of spurious and core features (L238–241).
> - Differences between LaSAR and retraining methods like AFR and JTT are elaborated (L133–134, L344–346, Section A.3), emphasizing LaSAR’s novelty in dimension-level spurious bias control (L87–88, L91–92), particularly in unsupervised settings (L59–60, L92–95).
>
> All major changes are highlighted in blue for ease of review. These updates aim to further clarify our contributions and underscore LaSAR’s theoretical and practical strengths.
>
> We remain committed to refining the manuscript to ensure clarity and impact. In light of the substantial improvements already made, we kindly request that you reconsider your score to reflect the progress in the latest version.
>
> Thank you again for your valuable feedback, which has been instrumental in strengthening our work.

---

### Official Review · Reviewer_rQEc · 2024-11-04

**Soundness:** 1
**Presentation:** 3
**Contribution:** 1
**Rating:** 5
**Confidence:** 4

**Summary:**

This paper addresses the issue of spurious correlations in deep neural networks trained with empirical risk minimization (ERM). The authors propose an approach called Last-Layer Selective Activation Retraining (LaSAR), which aims to mitigate spurious bias without requiring group labels or external annotations. The method selectively blocks neurons identified as spurious during the retraining of the last classification layer, thus promoting the model to learn robust decision rules. The authors demonstrate that LaSAR is effective across multiple data modalities, such as vision and text, and improves worst-group accuracy in benchmark datasets.

**Strengths:**

The proposed LaSAR method aims to achieve robust learning without group information by proposing metrics to evaluate whether a neuron is spurious or core related. This approach makes LaSAR a practical and fully unsupervised solution to mitigating spurious bias.

**Weaknesses:**

- Limited Theoretical Analysis: While the empirical results are promising, the theoretical foundation for why the proposed spuriousness score works effectively in all cases is very limited. Including more rigorous analysis or theoretical guarantees would strengthen the paper's claims about the effectiveness of LaSAR.
- Limited Heuristic Exploration: There is limited heuristic exploration of the distribution of the proposed spuriousness score. Figure 4 appears to be cherry-picked, and it would be more persuasive if the authors could provide the distribution of the proposed spuriousness score across neurons in different datasets.
- Incremental Contribution: The phenomenon that spurious neurons and core neurons can be separated has been demonstrated in prior work [1][2]. Moreover, the proposed spuriousness score is calculated as the median among misclassified samples and the median among correctly classified samples, which appears equivalent to retraining the last layer while up-weighting the incorrect samples. This limits the novelty of the contribution. Furthermore, the neuron masking algorithm assumes that a neuron can represent part of the spurious features, which is a strong assumption that may not always hold true. Additionally, it is unclear why masking the last layer is necessarily better than masking a middle layer.
- JTT Algorithm Classification: JTT is listed as a semi-supervised algorithm at line 362, but it appears to work without group information. This classification should be corrected.

[1] Last Layer Re-Training is Sufficient for Robustness to Spurious Correlations
[2] Complexity Matters: Dynamics of Feature Learning in the Presence of Spurious Correlations

**Questions:**

Distribution of Spuriousness Scores: Could the authors show the distribution of the proposed spuriousness scores across neurons in different datasets? This would help validate the claim that the spurious and core neurons can be effectively separated.

Difference from Retraining with Up-Weighting: What is the difference between the proposed algorithm and retraining the last layer while up-weighting the misclassified samples? Clarifying this would help in understanding the distinct contribution of the proposed method.

---

> ### Author Response · Authors · 2024-11-25
>
> 1. **Including more rigorous analysis or theoretical guarantees would strengthen the paper's claims about the effectiveness of LaSAR.**
>
>     Thanks for the suggestion. We have provided a theoretical analysis in the reivsed paper (Section 3.3, highlighted in blue color) with proofs provided in Appendix. In summary, we first gave  in **Proposition 1** the principal for selective activation, which is the core of LaSAR. Proposition 1 specifies the property of a neuron that is affected by spurious bias. Specifically, our selection method does not select neurons that exclusively represent a spurious or a core feature; instead, we select neurons that have high activation values but not for the target. Then, we theoretically showed in **Theorem 1** that our proposed spuriousness score indeed selects neurons in the last layer in a way that follows the proposed principal in Proposition 1. In other words, the score in general can effectively indentify neurons affected by spurious bias. Finally, we proved in **Theorem 2** that LaSAR can mitigate spurious bias in a model by brining the model closer to the unbiased one in the parametric space.
>
>
> 2. **Provide the distribution of the proposed spuriousness score across neurons in different datasets.**
>
>     Thanks for the suggestion. We have added additional visiualizations in Fig. 5 to Fig. 8 in Appendix to further support the finding that distributions of activation values provide useful information for identifying neurons affected by spurious bias (which is also theoretically proved in Theorem 1). The samples used for visualization are selected based on their activation values on the identified neurons, and we selected five samples that have the largest activation values on each identified neuron. In this way, we select samples that are mostly representative of the feature which the neuron represents.
>
>
> 3. **Regarding the contribution**
>
>     We hope to clarify that our contribution does not focus on demonstrating ``the phenomenon that spurious neurons and core neurons can be separated". Indeed, the prior work offers insights into spurious bias mitigation in the latent representations. Instead of demonstration, our contribution in this paper is to propose an automatic method that can explicitly identify individual neurons affected by spurious bias, which is different from prior works where this can only be achieved after retraining the model via bias mitigation methods, such as balanced sampling. Moreover, our method is fundamentally different from retraining the last layer while up-weighting the incorrect samples. Please kindly refer to the discussion in our answer to your question on this issue (Point 7). Finally, thanks to your suggestion, we have added a theoretical analysis to elucidate the principal of our method and validate our design choices.
>
> 4. **The neuron masking algorithm assumes that a neuron can represent part of the spurious features, which is a strong assumption that may not always hold true.**
>
>    Thanks for raising this concern. We hope to clarify that we do not make the assumption when using the neuron masking algorithm. Indeed, before this step, in Section 3.2.1, we have an additional step to identify neurons that are affected by spurious bias, i.e., neurons that mainly represent spurious features. Therefore, we ensure that the neuron masking algorithm masks out the correct neurons. Additionally, as shown in our theoretical analysis, the step in Section 3.2.1 indeed selects neurons with a strong reliance on spurious features. Empirically, the heatmaps provided in Fig. 5 to Fig. 8 support that a neuron can represent part of spurious features in a biased model.
>
>
> 6. **It is unclear why masking the last layer is necessarily better than masking a middle layer.**
>
>
>      Thanks for bringing this up. Indeed, we did not attempt to make the conclusion that masking the last layer is the optimal way of retraining. We made the first attempt to mask out neurons before the last layer for spurious bias mitigation and proved both empirically and theoretically that our approach is effective. We also emphasized in the revised paper that this approach is efficient. Masking neurons in a middle layer brings more freedom in designing methods for spurious bias mitigation, but this would also complicate the analysis and improve computational complexity. We greatly appreciate your insight and recognize it as a promising direction for future work.
>
>
> 6. **JTT Algorithm Classification**
>
>     Thanks for raising this concern. JTT does not need group information in the *training*, but it does require group information in the validation data for achieving optimal performance, which satisfies our definition of semi-supervised spurious bias mitigation. We have clarified this point in the related work section.

---

> > ### Author Response · Authors · 2024-11-25
> >
> > 7.  **What is the difference between the proposed algorithm and retraining the last layer while up-weighting the misclassified samples?**
> >
> >     Thanks for the question. Our method is fundamentally different from retraining the last layer while up-weighting the incorrect samples. First, our method only uses incorrectly predicted samples for identifying spurious neurons. If a neuron is masked out, then this will affect all the samples, not only the incorrect ones. In other words, different samples are treated in the same way in our framework with no preference to specific samples. Therefore, the equivalence between our method and the up-weighting one does not hold. Moreover, our method of mitigating spurious bias works in latent representations and has fine-grained control on spurious bias mitigation.  In contrast, the up-weighting method works at the sample-level and does not manipulate individual latent embedding dimensions.

---

> ### Comment · Reviewer_rQEc · 2024-12-03
>
> I appreciate the authors’ response and the additional experiments. I have raised my score by 1 level. However, my main concern remains regarding Comment 7. The authors explain how LaSAR operates differently from JTT, which I acknowledge and agree with. However, my point is that LaSAR’s approach of excluding neurons based on correct and incorrect samples is quite similar to retraining the last layer while up-weighting misclassified samples. While I understand that LaSAR allows for controlling embedding dimensions, this does not seem to be a critical difference to me.

---

> > ### Author Response · Authors · 2024-12-03
> >
> > Thank you for your thoughtful feedback and for raising your score. We appreciate the opportunity to clarify our approach further.
> >
> > - First, LaSAR operates in the **unsupervised spurious bias mitigation setting**, as outlined in Line 157, whereas JTT and similar retraining methods are designed for **semi-supervised settings** where group annotations are available. This distinction is fundamental because LaSAR achieves spurious bias mitigation **without requiring group labels**, making it applicable to broader scenarios where such annotations are costly or unavailable.
> >
> > - Second, LaSAR **directly zeroes out dimensions identified as spurious**, effectively removing their influence during retraining. In contrast, last-layer retraining methods indirectly address spurious bias by adjusting sample weights, which does not explicitly target embedding dimensions. This difference in methodology enables LaSAR to achieve finer-grained control over the representational space, which is critical in mitigating spurious correlations, as discussed in Lines 344-346 and Section A.3 of the Appendix.
> >
> > - Additionally, our theoretical analysis in Section 3.3 provides formal guarantees on the effectiveness of LaSAR's dimension-level intervention, highlighting its ability to identify and mitigate spurious dimensions more robustly compared to sample-based approaches. The experiments further validate that this dimension-centric approach yields superior robustness to spurious correlations.
> >
> > We hope this clarification addresses your concerns and highlights the unique contributions of LaSAR. We would greatly appreciate it if you could reconsider your score in light of these points. Thank you for your valuable feedback, which has significantly improved the quality and clarity of our work.

---

### Author Response · Authors · 2024-12-02
**Summary of Changes in the Revised Manuscript**

We sincerely thank the reviewers for their constructive feedback, which has helped us significantly improve the clarity, scope, and rigor of the manuscript. Below, we summarize the key updates made in response to the comments, with the goal of addressing concerns and demonstrating the contributions of our work:

- **Theoretical Analysis (rQEc, pwXU, fuS4):**

    We added a new theoretical analysis in Section 3.3 to demonstrate LaSAR’s effectiveness in identifying spurious dimensions and mitigating their impact. The analysis considers realistic scenarios where spurious features are distributed across multiple dimensions, providing formal evidence of LaSAR’s effectiveness in mitigating spurious bias.

- **New Experiments on ImageNet-9 and ImageNet-A (fuS4):**

    To address concerns about generalizability, we conducted additional experiments on larger and more challenging datasets, ImageNet-9 and ImageNet-A.  These experiments demonstrate LaSAR’s robust performance in handling distributional shifts and further validate its scalability and applicability to real-world tasks.

- **Discussion on Variable Selection (Ymhg):**

    We added a detailed discussion comparing LaSAR with traditional variable selection methods, highlighting its unique ability to handle spurious bias, which is often overlooked by conventional approaches. LaSAR’s unsupervised operation without group annotations makes it particularly useful in scenarios where annotated data is scarce.

- **Clarified Contributions (rQEc, pwXU):**

    We expanded our discussion on the similarities and differences between LaSAR and AFR (and other reweighting-based methods).
    We emphasized LaSAR’s novel dimensional-level approach, where spuriousness scores are used to mask biased dimensions directly, unlike AFR’s sample-level reweighting strategy. This distinction underscores LaSAR’s targeted mechanism for mitigating spurious bias.

- **Clarified Assumptions on Spurious and Core Features (fuS4, Qpa6):**

    We acknowledged that spurious features can span multiple neurons and provided evidence, both theoretical and experimental, to show that LaSAR is effective even under this realistic setting. We also refined our definition on spurious and core dimensions. This directly addresses concerns about the practicality of our assumptions.

We have revised sections throughout the paper to improve clarity and address reviewer concerns.
Additional details and discussions, including extended experiments and theoretical insights, are incorporated into both the main text and the Appendix.

We believe these substantial updates directly address the reviewers’ concerns, strengthening the theoretical foundations, broadening the empirical validation, and improving the clarity of our contributions.

We respectfully encourage the reviewers to reconsider their scores to reflect the substantial improvements made in response to their feedback. We deeply value the reviewers' insights and assure them that we will continue refining our paper based on their feedback.

Sincerely,\
Authors of Submission 10275

---

### Meta-Review · Area_Chair_3AEx · 2024-12-09

**Metareview:**

Spurious correlation is one issue of using standard empirical risk minimization with deep network models. Previous methods require annotations on spurious correlations. This paper demonstrates that it is possible to identify neurons affected by spurious bias without external supervision such as group labels. The proposed spuriousness score measures the spuriousness of certain dimensions for predicting certain classes. It utilize the prediction performance of the sample (i.e., correct or incorrect prediction) and the latent embedding of the sample based on the feature extractor of the network. Then the paper proposes to selectively retrain the last prediction layer based on the non-spurious neurons. Theoretical analysis and empirical investigation shows the benefits of the proposed approach.

Strengths: the paper has both theoretical analysis and empirical results to support the proposed method. It is practical since it does not require additional annotation. Experiments show good results.

Weaknesses: Initially, there were concerns about lack of theoretical insights/analysis and lack of real-world datasets in experiments. Several reviewers were concerned with the similarity with previous methods such as AFR and JTT and felt the contribution is incremental. Another concern was the similarity between variable selection methods. There were also concerns about the assumptions that the paper is making, e.g., spurious features can be isolated to specific neurons.

With the rebuttal and during the discussions, the authors have added theoretical analysis which shows that their method can move a model closer to an unbiased solution. More experiments were provided based on ImageNet-9 and ImageNet-A. The updated paper clarifies the difference between variable selection methods. The authors point out that they do not make an assumption that a neuron exclusively represents a spurious feature.

It seems many of the initial concerns were addressed, but one remaining concern is the novelty over similar approaches. Even after discussions and clarifications of the novelty by the authors, several reviewers were concerned that the main ideas in the proposed method are very similar to previous approaches such as AFR and JTT. The reviewers agree that the details differ (sample-wise approach or neuron-wise approach), but assessed that the novelty/significance is limited. Another remaining concern is the motivation of the design of the method (or some underlying assumptions that motivate the method): the authors provide more discussions and point to their new theory to motivate the design of the proposed method. One reviewer was satisfied with the response, but another reviewer mentioned that the assumption is still not motivated well. The final scores of reviewers were 6,6,5,3,3, tending slightly to the negative.

Based on the reviews, rebuttal, discussions among reviewers/authors, and the reasoning above, I would like to recommend rejection this time.

I encourage the authors to resubmit in the future. Since many of the concerns were addressed during the rebuttal/discussion phases, the paper's quality has improved significantly. I would like to suggest to carefully discuss the assumptions that the proposed method may rely on (or assumptions that the proposed method doesn't rely on) and novelty over previous methods (especially emphasize that the problem setting is different-doesn't require labeled validation set).

**Additional Comments On Reviewer Discussion:**

I explained some of the main discussion and changes during the rebuttal/discussion periods in the metareview above. The most important discussion was about the novelty over similar approaches such as JTT and AFR. After the discussions, two reviewers improved their score. 3 reviewers provided a final negative score (3, 3, 5), and my metareview's weaknesses section is mostly based on those 3 reviews.

---

### Decision · Program_Chairs · 2025-01-22

Reject